

# Lattice Clifford fractons and their Chern-Simons-like theory

**Weslei B. Fontana**[1,2⋆]**, Pedro R. S. Gomes**[2†] **and Claudio Chamon**[1‡]

**1** Physics Department, Boston University, Boston, MA, 02215, USA
**2** Departamento de Física, Universidade Estadual de Londrina,
86057-970, Londrina, PR, Brasil

⋆ weslei@uel.br, † pedrogomes@uel.br, ‡ chamon@bu.edu

## Abstract

We use Dirac matrix representations of the Clifford algebra to build fracton models on the lattice and their effective Chern-Simons-like theory. As an example, we build lattice fractons in odd $D$ spatial dimensions and their $(D + 1)$ spacetime dimensional effective theory. The model possesses an anti-symmetric $K$ matrix resembling that of hierarchical quantum Hall states. The gauge charges are conserved in sub-dimensional manifolds which ensures the fractonic behavior. The construction extends to any lattice fracton model built from commuting projectors and with tensor products of spin-1/2 degrees of freedom at the sites.

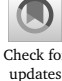
## Contents



# 1  Introduction

A major goal of condensed matter physics is to understand and to classify all possible phases of matter; another one is to uncover phases outside contemporary paradigms. While these two goals are evidently contradictory, together they move the field forward. An example of a new class of systems whose complete understanding is still in progress is that of what is now commonly referred to as fractons in general, or more precisely, systems with fracton excitations. These systems have peculiar properties, including ground state degeneracies that depend both on topology and geometry of lattice discretizations, and excitations with restricted mobility that, in turn, make the dynamical relaxation to the ground states slow [1–5].

Recent reviews of fractons can be found in [6] and [7]. Thus far, they are classified into two types: in Type I phases a single fracton excitation cannot move alone, but a pair can bind into mobile dipoles; in Type II phases, all excitations are immobile [3, 5, 8, 9]. It is this inherent immobility of isolated excitations that lead to slow dynamical behavior [1, 10, 11]. The same restricted mobility and slow dynamical relaxation of excitations might be useful for building quantum memories [12–14]. In addition, fractons possess connections to elasticity theory [15, 16] and gravity [17].

Fracton phases were originally constructed in lattice models; while their peculiar properties might appear unnatural for continuum descriptions, the construction of effective field theories that capture their low-energy properties is possible, as shown by Slagle and Kim [18] in the X-cube model [5, 19]. The construction of effective field theories enables much further progress [18, 20–25]. Some features of fracton excitations are captured by the field theories in simple ways. Restricted mobility, for example, is encoded in additional charge conservation laws along sub-dimensional manifolds, such as planes for 3-dimensional (3D)[1] models, besides the conservation of total charge in the whole volume. The conservation of charges in planes implies that a dipole in the perpendicular direction is conserved. Hence charge conservation in sub-manifolds is equivalent to the conservation of vector charges (dipoles), a feature of higher-rank gauge theories [26–39], which, in general, are gapless. Nevertheless, gapped fracton models can be obtained from higher-rank gauge theories via the Higgs mechanism [40, 41]. Gapped 3D fractons can also be obtained by either stacking [34, 42–47] or glueing [48–50] known (2 + 1)-dimensional topological orders.

You *et al.* present a different route to a fracton field theory that is not cast as a higher-rank gauge model. They present a Chern-Simons-like action with vector gauge fields that contains

---

[1]When referring to the dimensionality of the spacetime in this work we will use the notation $(D + 1)$, with $D$ the number of spatial components and the +1 refers to the time direction.

the sub-manifold conservation laws, hence also conserving dipoles. Their theory is gapped, and it can be discretized to a lattice to arrive at the Chamon model of Ref. [1]. The connection to Chern-Simons-like theories is appealing in that one would hope they can be generalized to describe classes of gapped fractons, much like Chern-Simons theories can describe classes of quantum Hall states [51].

In this work, we construct families of Chern-Simons-like theories of gapped fractons. These theories have multiple gauge charges, and are described by an anti-symmetric $K$ matrix and associated charge vectors. We arrive at these theories starting from microscopic lattice models, where we place a number $n$ of spin-1/2 degrees of freedom (or qubits) at the sites. Such starting point is rather generic, and encompasses models such as the Chamon and Haah codes. Instead of tensor products of Pauli operators, we use the Dirac representation of the Clifford algebra to describe the site degrees of freedom. We show that the Dirac representation with $2^n$-dimensional matrices is a natural mathematical framework to build the lattice models, and makes the connection to the field theory, a bosonization of sorts, rather simple. In the lattice theory, the fracton nature of the models are simple consequences of the lattice connectivity and the Clifford algebra, for example the immobility of single defects. In the continuum theory, these properties translate into charge conservation laws in sub-manifolds.

For the sake of giving a concrete but yet general example of the construction of these Clifford fractons, we build fracton models in any odd $D = 2n + 1$ spatial dimensions. This example allows one to track more easily the use of the $2^n \times 2^n$ anti-commuting Dirac matrices $\gamma^I$ with $I = 1, 2, \ldots, 2n + 1$, where $\gamma^1 \gamma^2 \ldots \gamma^{2n+1} = i^n$. The model of Ref. [1] corresponds to the simplest case, with $D = 3$ and $2 \times 2$ representations of the Dirac matrices. The $2n + 1$ Dirac matrices form a maximal set of anti-commuting operators, and no operator (any product of Dirac matrices) other than the identity commutes with less than two of the $\gamma$'s; it is this algebraic property that impedes the propagation of single fracton excitations.

We encode the anti-commutation relations of the Dirac matrices in a $2n \times 2n$ anti-symmetric matrix $K$ for constructing a model in the continuum. This "bosonization"-type scheme is a generalization of that in Ref. [18]. The generic bosonic formulation in terms of $K$ matrices and charge vectors allows us to take a continuum limit, and arrive at a (D+1)-dimensional Chern-Simons-like action

$$\mathcal{L} = \sum_{a,b=1}^{2n} \frac{1}{2\pi} K_{ab} A_a \partial_0 A_b + \frac{1}{\pi} \sum_{\alpha} K_{ab} A_0^{(\alpha)} \mathcal{D}_a^{(\alpha)} A_b \,, \tag{1}$$

where the differential $\mathcal{D}_a^{(\alpha)} = \sum_{I=1}^{D} T_a^{(I,\alpha)} \partial_I^2$ operators are tied to charge vectors $T^{(I,\alpha)}$ dictated by products of Dirac matrices in the microscopic lattice theory. The lattice model also determines the number of conserved currents that are minimally coupled to the $n$ fields $A_0^{(\alpha)}$, indexed by $\alpha = 1, \ldots, n$. The action is invariant under the $n$ gauge transformations

$$A_a \to A_a + \sum_{\alpha} \mathcal{D}_a^{(\alpha)} \zeta^{(\alpha)} \,, \tag{2}$$
$$A_0^{(\alpha)} \to A_0^{(\alpha)} + \partial_0 \zeta^{(\alpha)} \,,$$

if $K_{ab} \mathcal{D}_a^{(\alpha)} \mathcal{D}_b^{(\beta)} = 0$, again a condition ensured by relations between the microscopic lattice charge vectors $T^{(I,\alpha)}$ and the $K$ matrix. The conservation of the $n$ currents

$$\partial_0 J_0^{(\alpha)} = \mathcal{D}_a^{(\alpha)} J_a \,, \tag{3}$$

not only on the full volume, but also on sub-manifolds, follows from a linear dependence of the charge vectors, $\sum_{I=1}^{D} T^{(I,\alpha)} = 0$, which in turn affects the differential operators $\mathcal{D}_a^{(\alpha)}$.

These fracton models have a ground state degeneracy

$$\text{GSD} = \left[ 2^{\frac{D-1}{2}} \, \text{Pf}\,(K) \right]^{2^{D-2}\,L} , \tag{4}$$

for systems of linear size $L$ (hypervolume $L^D$). In the case of the "integer" fractons we constructed on the lattice, the $K$-matrix Pfaffian equals 1, and the degeneracy is $2^{(D-1)\,2^{D-3}\,L}$.

The paper is organized as follows. In Sec. 2, we construct, as an example, a microscopic Clifford fracton model in $D = 2n + 1$ spatial dimensions. In Sec. 3, we construct the corresponding effective theory in the continuum. In Sec. 4 we examine several properties of the effective field theories. We close in Sec. 5, with a brief summary and final remarks. Details of several computations as well as additional relevant discussions are presented in the appendices.

## 2 Microscopic Model in Arbitrary Odd Dimensions

The 3D model of Ref. [1] uses the simplest representation of the Clifford algebra, where the Dirac $\gamma$ matrices are $2 \times 2$: $\gamma^I = \sigma^I$, $I = 1, 2, 3$, with the $\sigma^I$ the Pauli matrices. The corresponding Hilbert space of the local degrees of freedom is 2-dimensional.

In 5D, for example, we can use the $4 \times 4$ representation of the Clifford algebra, with the 5 Dirac matrices $\gamma^I, I = 1, 2, 3, 4, 5$. (We work in Euclidean space, so we list the matrices from 1 to 4 plus the $\gamma^5$.) These matrices all anti-commute, $\{\gamma^I, \gamma^J\} = 2\delta_{IJ}$, and $\gamma^1\gamma^2\gamma^3\gamma^4\gamma^5 = i^2$. The local Hilbert space is 4-dimensional in this case. (This representation is obtained from a tensor product of two sets of Pauli matrices.) In $D = 2n + 1$ dimensions, we work with $2^n \times 2^n$ representations of the Clifford algebra, i.e. the Dirac matrices $\gamma^I, I = 1, \ldots, D$, satisfying $\prod_{I=1}^{D} \gamma^I = i^n$. (We build these matrices explicitly in appendix A.)

The construction of the fractons in odd-dimensional $D = 2n + 1$ space proceeds as follows. We start with an face-centered hypercubic lattice, that can be thought as the even sublattice $\Lambda_e$ of a hypercubic lattice with orthogonal basis vectors $\hat{a}_I, I = 1, \ldots, D$. We place the degrees of freedom on this even sublattice, as well as operators $\Gamma^{(I,\alpha)}$ with $\alpha = 1, \ldots, n$ acting on these degrees of freedom. The operators $\Gamma^{(I,\alpha)}$ are built as products of the $\gamma$-matrices (in turn built from tensor products of Pauli matrices, see appendix A). We take $\Gamma^{(I,1)} \equiv \gamma^I$, which we call principal configuration. The need for the additional $\Gamma^{(I,\alpha)}$ with $\alpha = 2, \ldots, n$ comes because the local Hilbert is $2^n$ dimensional, and consequently $n$ operators are necessary to gap the theory.

A generic $\Gamma$-operator can be parametrized in terms of a set of integer-valued vectors $T_a^{(I,\alpha)}$, $a = 1, \ldots, 2n$, according to

$$\Gamma^{(I,\alpha)} = \left(\gamma^1\right)^{T_1^{(I,\alpha)}} \left(\gamma^2\right)^{T_2^{(I,\alpha)}} \cdots \left(\gamma^{2n}\right)^{T_{2n}^{(I,\alpha)}} . \tag{5}$$

Furthermore, since $(\gamma^I)^2 = 1$, only the values of the $T$-vectors mod 2 matter. For the principal configuration we can choose, for example, the vectors

$$T_a^{(I,1)} \equiv t_a^{(I)} \equiv \delta_a^I, \ I = 1, \ldots, 2n, \quad \text{and} \quad T_a^{(2n+1,1)} \equiv t_a^{(2n+1)} \equiv -\sum_{I=1}^{2n} t_a^{(I)} . \tag{6}$$

We call this choice as the canonical form. Written explicitly, $t^{(1)} = (1, 0, \ldots, 0)$, $t^{(2)} = (0, 1, \ldots, 0), \ldots, t^{(2n)} = (0, 0, \ldots, 1)$ and $t^{(2n+1)} = (-1, -1, \ldots, -1)$. The condition $\sum_{I=1}^{D} t_a^{(I)} = 0$ is tied to the fact that all the $\gamma^I$ multiply to the identity (up to a phase).

We define $\mathcal{O}^{(\alpha)}$ operators centered on the odd sublattice $\Lambda_o$,

$$\mathcal{O}_{\vec{x}}^{(\alpha)} \equiv \prod_{I=1}^{D} \Gamma_{\vec{x}-\hat{a}_I}^{(I,\alpha)} \, \Gamma_{\vec{x}+\hat{a}_I}^{(I,\alpha)}, \quad \alpha = 1, \dots, \frac{(D-1)}{2} \, . \tag{7}$$

Notice that $\left(\mathcal{O}_{\vec{x}}^{(\alpha)}\right)^2 = \mathbb{1}$ follows because the $\Gamma^{(I,\alpha)}$ are products of Dirac matrices. Using these operators we construct the Hamiltonian

$$H = -\sum_{\alpha=1}^{(D-1)/2} \left( g_\alpha \sum_{\vec{x}} \mathcal{O}_{\vec{x}}^{(\alpha)} \right), \tag{8}$$

where all coupling constants $g_\alpha$ are chosen to be positive. We can further choose the operators $\Gamma^{(I,\alpha)}$ such that

$$\left[ \mathcal{O}_{\vec{x}}^{(\alpha)}, \, \mathcal{O}_{\vec{x}'}^{(\beta)} \right] = 0, \quad \forall \, \alpha, \beta \quad \text{and} \quad \forall \, \vec{x}, \vec{x}' \, . \tag{9}$$

In this case, the Hamiltonian is a sum of commuting projectors and there are as many commuting projectors (up to constraints that we shall see in a moment give a topological degeneracy) as the number of degrees of freedom in the problem.

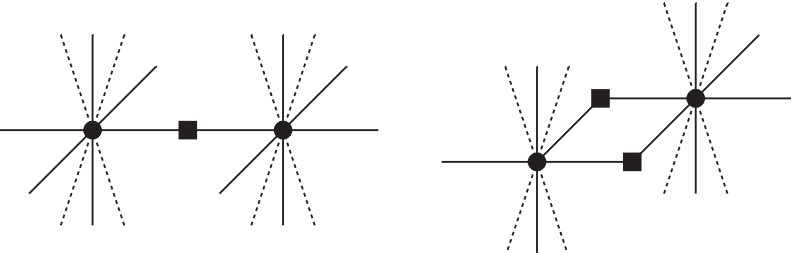

Figure 1: The two possibilities for distinct operators $\mathcal{O}$ sharing sites. The black squares correspond to the sites of the even sublattice $\Lambda_e$, while black dots correspond to the sites of the odd sublattice $\Lambda_o$. The dotted lines represent additional dimensions.

The connection between the choice of operators $\Gamma^{(I,\alpha)}$ and the commutations between the $\mathcal{O}_{\vec{x}}^{(\alpha)}$ stems from the geometry imprinted via the definition Eq. (7), and is depicted in Fig. 1. The $\mathcal{O}^{(\alpha)}$'s trivially commute both when they are defined at the same site $\vec{x} \in \Lambda_o$ or when they do not share any sites (in $\Lambda_e$); there just remains two cases to be checked: when they share one and two sites. The neighboring $\mathcal{O}^{(\alpha)}$'s, defined at sites $\vec{x}$ and $\vec{x} + 2\hat{a}_I$ of $\Lambda_o$, share the $\Lambda_e$ site at $\vec{x} + \hat{a}_I$, and they commute if

$$\left[ \Gamma^{(I,\alpha)}, \, \Gamma^{(I,\beta)} \right] = 0 \, . \tag{10}$$

Neighboring $\mathcal{O}^{(\alpha)}$'s, defined at sites $\vec{x}$ and $\vec{x} + \hat{a}_I + \hat{a}_J$ of $\Lambda_o$, share the two $\Lambda_e$ sites at $\vec{x} + \hat{a}_I$ and $\vec{x} + \hat{a}_J$. The operators on those sites either commute or anti-commute, which can be cast as

$$\Gamma^{(I,\alpha)} \, \Gamma^{(J,\beta)} = (-1)^{\eta_{IJ}^{(\alpha\beta)}} \, \Gamma^{(J,\beta)} \, \Gamma^{(I,\alpha)} \, , \tag{11}$$

with $\eta_{IJ}^{(\alpha\beta)} = 0$ or 1. The desired commutation relations Eq. (9) are guaranteed if

$$\eta_{IJ}^{(\alpha\beta)} = \eta_{JI}^{(\alpha\beta)} \, . \tag{12}$$

In particular, the commutation (10) implies $\eta_{II}^{(\alpha,\beta)} = 0$.

All these conditions can be satisfied using Dirac matrix representations of the Clifford algebra. The simplest example is the $D = 3$ contained in Ref. [1], where one uses the $2 \times 2$ representation

$$
\begin{array}{c|ccc}
\Gamma^{(I,\alpha)} & I = & 1 & 2 & 3 \\
\hline
\alpha = 1 & & \gamma^1 = \sigma^1 & \gamma^2 = \sigma^2 & \gamma^3 = \sigma^3
\end{array}
\tag{13}
$$

In $D = 5$ we use the 4-dimensional representation of the Dirac matrices and take the following $\Gamma^{(I,\alpha)}$ operators:

$$
\begin{array}{c|ccccc}
\Gamma^{(I,\alpha)} & I = & 1 & 2 & 3 & 4 & 5 \\
\hline
\alpha = 1 & & \gamma^1 & \gamma^2 & \gamma^3 & \gamma^4 & \gamma^5 \\
\alpha = 2 & & \gamma^3\gamma^5 & \gamma^4\gamma^5 & \gamma^1\gamma^5 & \gamma^2\gamma^5 & \gamma^5
\end{array}
\tag{14}
$$

that satisfy Eqs. (10) and (12) and hence yield the set of commuting projectors $\mathcal{O}^{(1)}_{\vec{x}}$ and $\mathcal{O}^{(2)}_{\vec{x}}$, $\vec{x} \in \Lambda_o$. As we shall see in the next section, the field theory formulation provides a systematic way to construct operators of the sets $\alpha \geq 2$ in arbitrary odd dimensions and satisfying all the required commutation rules.

The ground state of these models (for any $D = 2n + 1$) correspond to all $\mathcal{O}^{(\alpha)}_{\vec{x}}$ having eigenvalue $+1$. Excitations or defects correspond to those operators having instead eigenvalue $-1$. That these models are fractonic requires that there is not a single local operator whose net effect is to generate a defect pair or equivalently move a single isolated defect. In $D = 5$, for example, one can easily check that there is no product of Dirac operators that anti-commutes with a single $\Gamma^{(I,\alpha)}$ operator for given $\alpha$. The minimum number of defects that can be created or anihilated is four, like in the $D = 3$ model of Ref. [1]; this number four remains the same for any odd dimension $D$ model. (See appendix B for details.)

A lower bound to the topological ground state degeneracy of the model can be placed by noticing that the $\mathcal{O}^{(\alpha)}_{\vec{x}}$ have the property

$$
\prod_{\vec{x} \in \Lambda_{o,k}} \mathcal{O}^{(\alpha)}_{\vec{x}} = \mathbb{1}, \quad k = 1, \ldots, 2^{D-1}, \quad \alpha = 1, \ldots, (D-1)/2,
\tag{15}
$$

where $k$ accounts for all the $2^{D-1}$ sub-lattices $\Lambda_{o,k}$. Eq.(15) already give us a hint of the degeneracy of the model, which will be at least $2^{[(D-1)\,2^{D-2}]}$, but it happens that this number is a lower bound to the degeneracy and in fact, the degeneracy can also depend on the system size, as was shown in [2]. For a hypercube of volume $L^D$, the degeneracy dependence on the linear size $L$ is $2^{(D-1)\,2^{D-3}\,L}$ (see appendix C for details).

## 3  Low-Energy Effective Field Theory

In this section we shall derive an effective field theory capturing the low-energy physical properties of the lattice models given by (8). To connect the operators that act on the microscopic degrees of freedom with the suitable operators that possess a well-defined continuum limit, we define a map parametrized by the vectors $T^{(I,\alpha)}_a$ of Eq. (5):

$$
\Gamma^{(I,\alpha)}_{\vec{x}} \equiv \exp\left(i\,T^{(I,\alpha)}_a K_{ab} A_b(\vec{x})\right),
\tag{16}
$$

where the repeated matrix indices $a, b = 1, \ldots, 2n$ are summed. We need only $D - 1 = 2n$ independent fields $A_a$ to construct all the required operators. In addition, we see that this parametrization introduces a symmetry

$$
A \to QA, \quad T^{(I,\alpha)} \to Q\,T^{(I,\alpha)}, \quad K \to \left(Q^\top\right)^{-1} K\,Q^{-1},
\tag{17}
$$

with $Q$ an arbitrary matrix.

For the case of the principal configuration, where $\Gamma^{(I,1)} \equiv \gamma^I$ and, accordingly, $T_a^{(I,1)} \equiv t_a^{(I)}$, we have

$$\Gamma_{\vec{x}}^{(I,1)} \equiv \gamma_{\vec{x}}^I \equiv \exp\left(i\, t_a^{(I)} K_{ab} A_b(\vec{x})\right). \tag{18}$$

Properties of the fields $A_a$ and of the matrix $K$ can be obtained from the analysis of the principal configuration. Indeed, we start by computing

$$\gamma_{\vec{x}}^I\, \gamma_{\vec{x}'}^J = \exp\left(-\left[t_a^{(I)} K_{ab} A_b(\vec{x}), \, t_{a'}^{(J)} K_{a'b'} A_{b'}(\vec{x}')\right]\right) \gamma_{\vec{x}'}^J\, \gamma_{\vec{x}}^I, \tag{19}$$

where we have used the BCH formula and assumed that the commutator appearing in this expression is a $c$-number. Since $\gamma_{\vec{x}}^I$ and $\gamma_{\vec{x}'}^J$ must commute if $\vec{x} \neq \vec{x}'$ and anti-commute if $\vec{x} = \vec{x}'$ and $I \neq J$, we impose

$$\left[A_b(\vec{x}), A_{b'}(\vec{x}')\right] \equiv i\pi (K^{-1})_{bb'}\, \delta_{\vec{x}\vec{x}'}, \tag{20}$$

which will be interpreted as an equal-time commutation relation in a field theory formulation with a canonical pair $A_a(\vec{x})$ and $\Pi_a(\vec{x}) = \frac{1}{\pi} (K^T)_{ab} A_b$. We shall return to this point later. Using the commutation relation (20) in (19) leads to the further following conditions to match the anti-commutation relations among the $\gamma_{\vec{x}}^I$:

$$t_a^{(I)} (K^\top)_{ab}\, t_b^{(J)} = \begin{cases} 1 \mod 2, & I \neq J \\ 0 \mod 2, & I = J. \end{cases} \tag{21}$$

The condition (21) is particular to the principal configuration, and reflects that the building blocks of the theory are anti-commuting objects (Dirac matrices); it is not needed for the other $T$-vectors with $\alpha \geq 2$, since the associated operators (products of Dirac matrices) may either commute or anti-commute. Notice that the conditions in (21) imply that the fields are compact, since the shifts

$$A_b \rightarrow A_b + 2\pi \sum_{J=1}^{D} t_b^{(J)} m_J, \quad m_J \in \mathbb{Z}, \tag{22}$$

do not change $\gamma_{\vec{x}}^I$ because

$$\exp\left(2i\pi \sum_{J=1}^{D} t_a^{(I)} K_{ab} t_b^{(J)} m_J\right) = 1. \tag{23}$$

The most general solution (independent of $t$) for the condition in the second line of (21) corresponds to the case where $K$ is an anti-symmetric matrix. In writing the commutation relation (20) we assumed that the inverse $K^{-1}$ exists, which requires $\det K \neq 0$, a condition only possible to satisfy for even-dimensional anti-symmetric $K$ matrices. Recall that $K$ is a $2n \times 2n = D-1 \times D-1$ matrix, so the construction works for odd dimensions $D$.

A useful relation involving the $t$-vectors emerges when we consider the product of all matrices $\gamma$ in the same site. Suppressing the matrix indices for simplicity, we have

$$\begin{aligned}
\mathbb{1} \sim \gamma^1 \gamma^2 \cdots \gamma^D &= e^{it^{(1)}KA} e^{it^{(2)}KA} \cdots e^{it^{(D)}KA} \\
&= \exp\left[i \sum_{I=1}^{D} t^{(I)} KA\right] \exp\left[\frac{\pi i}{2} \sum_{I<J} t^{(I)} K t^{(J)}\right] \\
&= \exp\left[i \sum_{I=1}^{D} t^{(I)} KA\right] \exp\left[\frac{\pi i}{2} \frac{(D-1)D}{2}\right], \tag{24}
\end{aligned}$$

where we have used (21) in the last step. In order for the right-hand side to be proportional to the identity, we require

$$\sum_{I=1}^{D} t_a^{(I)} = 0 \,, \tag{25}$$

which we refer to as the neutrality condition. Notice that this is satisfied with the choice in (6). Moreover, we shall require the neutrality condition for all sets of operators $\Gamma^{(I,\alpha)}$, which corresponds to

$$\sum_{I=1}^{D} T_a^{(I,\alpha)} = 0 \,. \tag{26}$$

We now proceed by analyzing the field theory counterpart of the lattice operators (7). Using the representation (16), it follows that

$$\mathcal{O}_{\vec{x}}^{(\alpha)} = \exp\left[ i \sum_{J=1}^{D} \left( T_a^{(J,\alpha)} K_{ab} A_b(\vec{x} + \hat{a}_J) + T_a^{(J,\alpha)} K_{ab} A_b(\vec{x} - \hat{a}_J) \right) \right]. \tag{27}$$

Given the commutation relation (20), we have to determine the conditions on $T^{(J,\alpha)}$ and $K$ so as to produce commuting operators $\mathcal{O}_{\vec{x}}^{(\alpha)}$, i.e., so that the field theory representation on the right-hand side reproduces the commutations in Eq. (9). As discussed in the previous section, there are two situations where a nontrivial commutation rule may arise: when the operators share one or two sites. When they do not share any site, the commutation is trivially satisfied. To take into account these two situations, we just have to consider two operators $\mathcal{O}_{\vec{x}}^{(\alpha)}$ at positions $\vec{x}$ and $\vec{x} + \vec{a}_I + \vec{a}_J$. Thus, if $I = J$ the operators share the site $\vec{x} + \vec{a}_I$ and if $I \neq J$ they share the two sites $\vec{x} + \vec{a}_I$ and $\vec{x} + \vec{a}_J$. The requirement of commutation between the operators $\mathcal{O}_{\vec{x}}^{(\alpha)}$ and $\mathcal{O}_{\vec{x}+\vec{a}_I+\vec{a}_J}^{(\beta)}$ is

$$C_{IJ}^{(\alpha\beta)} = 0 \,, \tag{28}$$

where

$$C_{IJ}^{(\alpha\beta)} \equiv T_a^{(I,\alpha)} K_{ab} T_b^{(J,\beta)} + T_a^{(J,\alpha)} K_{ab} T_b^{(I,\beta)} \,. \tag{29}$$

Notice that $C_{IJ}^{(\alpha\beta)} = C_{JI}^{(\alpha\beta)}$ and $C_{IJ}^{(\alpha\beta)} = -C_{IJ}^{(\beta\alpha)}$. The symmetry in the $IJ$ indices follows directly from the way $C_{IJ}^{(\alpha\beta)}$ is defined, whereas the anti-symmetry in the $\alpha\beta$ indices follows from the anti-symmetry of the matrix $K$. In particular, the condition (28) is automatically satisfied if $\alpha = \beta$, which is consistent with the fact that $\mathcal{O}_{\vec{x}}^{(\alpha)}$ operators of the same kind commute with each other. A systematic procedure for constructing $T$-vectors satisfying the condition (28) is presented in appendix B. Next we consider the continuum limit of the relation (27). The expansion of the field $A_b$ reads

$$A_b(\vec{x} \pm \hat{a}_J) = A_b(\vec{x}) \pm \sum_{I} a_J^I \, \partial_I A_b(\vec{x}) + \frac{1}{2} \sum_{I,K} a_J^I a_J^K \, \partial_I \partial_K A_b(\vec{x}) + \cdots . \tag{30}$$

As the unit vectors $\hat{a}_J$ have the components $a_J^I = \delta_J^I$, we get

$$\mathcal{O}_{\vec{x}}^{(\alpha)} = \exp\left( 2i \sum_{J=1}^{D} T_a^{(J,\alpha)} K_{ab} A_b(\vec{x}) + i \sum_{J=1}^{D} T_a^{(J,\alpha)} K_{ab} \partial_J^2 A_b(\vec{x}) + \dots \right). \tag{31}$$

We see that the neutrality condition (26) ensures that the first term in the exponential vanishes, so that the operator $\mathcal{O}_{\vec{x}}^{(\alpha)}$ reduces to

$$\mathcal{O}_{\vec{x}}^{(\alpha)} = \exp\left( i \sum_{J=1}^{D} T_a^{(J,\alpha)} K_{ab} \partial_J^2 A_b(\vec{x}) + \cdots \right). \tag{32}$$

The Hamiltonian in (8) becomes

$$H \sim -2 \sum_\alpha g_\alpha \int d^D x \, \cos\left(M^{(\alpha)}(\vec{x})\right), \tag{33}$$

with

$$M^{(\alpha)}(\vec{x}) \equiv \sum_{J=1}^{D} T_a^{(J,\alpha)} K_{ab} \, \partial_J^2 A_b(\vec{x}). \tag{34}$$

We see that the ground state corresponds to the case where all the cosines in (33) are simultaneously pinned at $M^{(\alpha)} = 2\pi m^{(\alpha)}$ for all sites, where $m^{(\alpha)} \in \mathbb{Z}$. We can enforce this in a corresponding field theory description of the ground state through a Lagrange multiplier, as we will discuss in a moment.

Before going to the field theory it is convenient to express the operator $M^{(\alpha)}(\vec{x})$ in a way that solves the constraint of the neutrality condition (26). Thus, we single out one of the directions, say the last one $J = D$, and write

$$
\begin{aligned}
M^{(\alpha)}(\vec{x}) &= \sum_{J=1}^{D-1} T_a^{(J,\alpha)} K_{ab} \, \partial_J^2 A_b(\vec{x}) + T_a^{(D,\alpha)} K_{ab} \, \partial_D^2 A_b(\vec{x}) \\
&= \sum_{J=1}^{D-1} T_a^{(J,\alpha)} K_{ab} \, D_J A_b(\vec{x}),
\end{aligned}
\tag{35}
$$

where the derivative operator $D_J$ is defined as $D_J \equiv \partial_J^2 - \partial_D^2$. It is also convenient to define another differential operator as

$$
\begin{aligned}
\mathcal{D}_a^{(\alpha)} &\equiv \sum_{J=1}^{D} T_a^{(J,\alpha)} \partial_J^2 \\
&= \sum_{J=1}^{D-1} T_a^{(J,\alpha)} D_J.
\end{aligned}
\tag{36}
$$

In terms of $\mathcal{D}_a^{(\alpha)}$, the operator $M^{(\alpha)}(\vec{x})$ in (35) acquires a simple compact form

$$M^{(\alpha)}(\vec{x}) = K_{ab} \, \mathcal{D}_a^{(\alpha)} A_b, \tag{37}$$

which makes evident its invariance under gauge transformations

$$A_a \rightarrow A_a + \sum_\alpha \mathcal{D}_a^{(\alpha)} \zeta^{(\alpha)}, \tag{38}$$

with $\zeta^{(\alpha)} = \zeta^{(\alpha)}(\vec{x}, t)$ being a set of arbitrary functions of spacetime coordinates. In fact, notice that

$$
\begin{aligned}
K_{ab} \, \mathcal{D}_a^{(\alpha)} \mathcal{D}_b^{(\beta)} &= \sum_{I,J=1}^{D} K_{ab} \, T_a^{(I,\alpha)} T_b^{(J,\beta)} \, \partial_I^2 \, \partial_J^2 \\
&= \sum_{I,J=1}^{D} C_{IJ}^{(\alpha\beta)} \, \partial_I^2 \, \partial_J^2 \\
&= 0, \quad \text{since } C_{IJ}^{(\alpha\beta)} = 0.
\end{aligned}
\tag{39}
$$

Therefore, the condition above, needed for gauge invariance, is precisely the condition for commutation of the cosine operators (28).

With all these elements in place, we can write down a field theory which describes the ground state of the microscopic fracton model,

$$S = \int d^D x \, dt \, \frac{1}{2\pi} \left[ K_{ab} \, A_a \, \partial_0 A_b + 2 \sum_\alpha A_0^{(\alpha)} \, K_{ab} \, \mathcal{D}_a^{(\alpha)} \, A_b \right] . \tag{40}$$

The first term is responsible for the commutation relation (20)[2], whereas the second one enforces the ground state constraints, with $A_0^{(\alpha)}$ a set of Lagrange multipliers. The requirement of full gauge invariance of the action (up to boundary terms) dictates that $A_0^{(\alpha)}$ must transform as

$$A_0^{(\alpha)} \to A_0^{(\alpha)} + \partial_0 \zeta^{(\alpha)} . \tag{41}$$

Thus, we end up with a *bona fide* gauge theory, which resembles the Chern-Simons description of topologically ordered systems. The gauge-invariant "electric" and "magnetic" fields can be defined as

$$E_a \equiv \partial_0 A_a - \sum_\alpha \mathcal{D}_a^{(\alpha)} A_0^{(\alpha)} \quad \text{and} \quad B_{a_1 a_2 \cdots a_{D-3}}^{(\alpha)} \equiv \epsilon_{a_1 a_2 \cdots a_{D-1}} \, \mathcal{D}_{a_{D-2}}^{(\alpha)} A_{a_{D-1}} , \tag{42}$$

where $\epsilon_{a_1 a_2 \cdots a_{D-1}}$ is the Levi-Civita tensor of rank $D-1$.

## 4 Properties of the Effective Theory

### 4.1 Level Quantization

Now we will explore some properties of the effective field theory (40). Firstly, it is interesting to understand whether there is a notion of quantization of the "level" of the theory, which in the present case is given by the matrix $K$. To address this question we consider the principal configuration $T^{(I,1)} = t^{(I)}$. In this case, the $t$-vectors must satisfy the conditions in (21). Then, we use the symmetry transformations in (17) to make a specific choice for the $t$-vectors. For example, if we pick up the canonical form (6), we obtain the following level quantization condition:

$$K_{IJ} = \text{odd} , \quad \text{with } I \neq J \quad \text{and} \quad I, J = 1, \ldots, D-1 , \tag{43}$$

i.e., all the off-diagonal elements must be odd integers, and consequently nonvanishing. Of course, different representations of the $t$-vectors yield different quantization of the elements of the matrix $K$, but in all the cases we end up with some notion of quantization due to the conditions in (21).

From the field theory alone, the quantization of the level can be understood as follows. Consider a manifold $M = \mathcal{S}^1 \times \mathcal{M}^D$, with $\mathcal{S}^1$ representing the time direction with period $[0, \tau)$ and $\mathcal{M}^D$ a spatially closed manifold. Due to the compact nature of the fields $A_a$, we have a quantized flux

$$\int_{\mathcal{M}^D} B_{a_1 a_2 \ldots a_{D-3}}^{(\alpha)} \equiv \pi \, p_{a_1 a_2 \ldots a_{D-3}}^{(\alpha)}, \quad p_{a_1 a_2 \ldots a_{D-3}}^{(\alpha)} \in \mathbb{Z} . \tag{44}$$

Consider large gauge transformations that wind around the time direction. The $A_0^{(\alpha)}$ field transform as

$$A_0^{(\alpha)} \to A_0^{(\alpha)} + \frac{2\pi}{\tau} n^{(\alpha)}, \quad n^{(\alpha)} \in \mathbb{Z}, \tag{45}$$

---

[2]Notice that the prefactor of $\frac{1}{2}$ in the action (40) ensures the right numerical factor in the commutation relation (20), since for each pair of coordinates we always have two contributions because of the anti-symmetry of the matrix $K$, for example, $K_{12} (A_1 \partial_0 A_2 - A_2 \partial_0 A_1)$. This pair of terms must be brought into a single term through integration by parts before computing the canonical momentum.

and the corresponding variation of the action under these transformations is

$$\delta S = \pi K_{ab} \sum_{\alpha} \frac{1}{(D-3)!} \, n^{(\alpha)} \, \epsilon_{a_1 a_2 \ldots a_{D-3} ab} \, p^{(\alpha)}_{a_1 a_2 \ldots a_{D-3}}$$
$$= \pi K_{ab} \, \mathbb{Z}_{ab} \,, \tag{46}$$

where $\mathbb{Z}_{ab}$ an integer valued quantity obtained from the summation above. From this, it is straightforward to note that in order for the quantum theory to be invariant under the large gauge transformations (45), the $K$-matrix elements have to be integer valued. Further details of this calculation are found in appendix D

## 4.2 Three dimensional case

It is instructive to compare the effective field theory that we have obtained for the particular case of three spatial dimensions, $D = 3$, with the result of Ref. [20]. For $D = 3$ there is only one configuration, the principal configuration $\alpha = 1$. The matrix $K$ in this case is

$$K = \begin{pmatrix} 0 & k \\ -k & 0 \end{pmatrix} \quad \text{and} \quad K^{-1} = \begin{pmatrix} 0 & -\frac{1}{k} \\ \frac{1}{k} & 0 \end{pmatrix} . \tag{47}$$

The action, in terms of electric and magnetic fields, reduces to

$$S = \int d^3 x \, dt \, \frac{k}{2\pi} \, [A_1 E_2 - A_2 E_1 + A_0 B], \quad [A_1(\vec{x}), A_2(\vec{x}')] = -\frac{\pi i}{k} \, \delta \left( \vec{x} - \vec{x}' \right) . \tag{48}$$

With the canonical choice (6), the derivative operators entering the electric and magnetic fields become $\mathcal{D}_1 = \partial_1^2 - \partial_3^2$ and $\mathcal{D}_2 = \partial_2^2 - \partial_3^2$, whereas the coefficient $k$ must be an odd integer, in accordance with (43). In order to compare with [20], we just need to rename the fields and the derivative operators according to $A_1 \to -A_2$, $A_2 \to A_1$, $\mathcal{D}_1 \to -\mathcal{D}_2$, and $\mathcal{D}_2 \to \mathcal{D}_1$ (see equation (95) of [20]), which leave both the action and the commutation relation in (48) unchanged. In this form, we can immediately compare with the results of [20] with the following identification between the parameters $k = s/2$. In that work, the original Chamon model (with full cubic symmetry) is recovered for $s = 2$ (in [20], the level quantization is $s \in \mathbb{Z}$), which in our normalization corresponds to $k = 1$. This choice describes the 2-state system at each site, as expected. Also, this choice of $k$ is allowed by the level quantization (43) associated with the canonical choice for the $t$-vectors. For a general discussion of how the $K$-matrix elements are determined by the microscopic theory, we refer the reader to appendix E.

## 4.3 Conservation Laws

A gauge-invariant coupling to matter can be introduced in the action (40) through the terms $\sum_{\alpha} A_0^{(\alpha)} J_0^{(\alpha)} + A_a J_a$, provided that the current satisfies the continuity equation

$$\partial_0 J_0^{(\alpha)} = \mathcal{D}_a^{(\alpha)} J_a \,. \tag{49}$$

By integrating over the whole space and assuming periodic boundary conditions along all directions, it follows that charge is conserved in the whole system,

$$\frac{d}{dt} \int d^D x \, J_0^{(\alpha)} = \int d^D x \, \mathcal{D}_a^{(\alpha)} J_a = 0 \,. \tag{50}$$

In addition, given the form of the derivative operators $\mathcal{D}_a^{(\alpha)}$, we also have more restrictive conservation laws. These extra conservation laws require that charge is also conserved on a

set of sub-manifolds of the system. It is due to these extra conservation laws that the fracton behavior of the excitations emerges.

To find the sub-manifolds where charge is conserved, we use the definition of the derivative operators $\mathcal{D}_a^{(\alpha)}$ in (36) to write the continuity equation as

$$
\begin{aligned}
\partial_0 J_0^{(\alpha)} &= \sum_{I=1}^{D-1} T_a^{(I,\alpha)} D_I J_a, \\
&= \sum_{I=1}^{D-1} D_I J_I^{(\alpha)},
\end{aligned}
\tag{51}
$$

where we have defined $J_I^{(\alpha)} \equiv T_a^{(I,\alpha)} J_a$. Recalling that $D_I \equiv \partial_I^2 - \partial_D^2$, it is convenient to introduced the directions $\hat{x}_{ID}^{\sigma_I} \equiv \hat{a}_I + \sigma_I \hat{a}_D$, with $\sigma_I = \pm 1$. In this notation, (51) can be written as

$$
\partial_0 J_0^{(\alpha)} = 4 \sum_{I=1}^{D-1} \left( \partial_{ID}^- \partial_{ID}^+ \right) J_I^{(\alpha)}.
\tag{52}
$$

This form of the continuity equation induces $2^{D-1}$ extra conservation laws, explicitly, that charge must be conserved in each of the $(D-1)$-dimensional sub-manifolds labelled by $\left( x_{1D}^{\sigma_1}, \ldots, x_{(D-1)D}^{\sigma_{D-1}} \right)$. Indeed, if we integrate $J_0^{(\alpha)}$ over any of these sub-dimensional manifolds, we obtain the following conserved charges

$$
Q_{(\sigma_1, \sigma_2, \ldots, \sigma_{D-1})}^{(\alpha)} \equiv \int dx_{1D}^{\sigma_1} dx_{2D}^{\sigma_2} \ldots dx_{(D-1)D}^{\sigma_{D-1}} J_0^{(\alpha)}.
\tag{53}
$$

These conservation laws, in turn, imply that the dipole moment in the direction perpendicular to those manifolds is conserved. Naturally, such conservation laws impose several restrictions on the mobility of the particles. We build in detail the form of the excitations in appendix F.

### 4.4 Ground State Degeneracy

Here we discuss the computation of the ground state degeneracy using the effective field theory. Naturally, in the continuum limit the degeneracy is infinite so that we shall adopt some kind of discretization (regularization) of the theory. The form of the conservation laws in the sub-dimensional manifolds provides a very natural way to discretize the theory in a layered structure. The basic idea is to consider the system as a stack of layers corresponding to the sub-dimensional manifolds where charge is conserved.

Let us start with the case $D = 3$. The action (40) reduces to

$$
S = \int d^3x \, dt \, \frac{k}{\pi} A_1 \, \partial_0 A_2 + \cdots,
\tag{54}
$$

where we keep explicitly only the part relevant for the computation of the degeneracy. In this case, charge is conserved in $2^2$ sub-spaces labeled by $\sigma_1, \sigma_2 = \pm$, with the corresponding measures

$$
\int dx_{13}^{\sigma_1} dx_{23}^{\sigma_2}.
\tag{55}
$$

The strategy is to write the action (54) in terms of the coordinates $x_{13}^{\sigma_1}, x_{23}^{\sigma_2}, x_\perp$, where $x_\perp$ is the coordinate perpendicular to the plane defined by the directions $x_{13}^{\sigma_1}$ and $x_{23}^{\sigma_2}$. Upon this change of variables,

$$
\int d^3x \to \int dx_{13}^{\sigma_1} dx_{23}^{\sigma_2} dx_\perp \, \mathcal{J},
\tag{56}
$$

where $\mathcal{J}$ is the Jacobian of the transformation. As this transformation is linear, $\mathcal{J}$ is just a constant and can be absorbed in $dx_\perp$. The transformation from the coordinates $x_1$, $x_2$, $x_3$ to $x_{13}^{\sigma_1}$, $x_{23}^{\sigma_2}$, $x_\perp$ will change the limits of integration. However, as the ground state degeneracy in each plane with periodic boundary conditions (forming a torus $T^2$) does not depend on the area of the plane (torus), so we can ignore the area of integration in our computation as long as we assume periodic boundary conditions along the plane $x_{13}^{\sigma_1}$-$x_{23}^{\sigma_2}$.

The next step is to discretize the coordinate $x_\perp$. We consider that the perpendicular direction is composed by a stack of $N$ layers,

$$\int dx_\perp \to \sum_{i=1}^{N} 2a \,, \tag{57}$$

where $2a$ is the separation between the planes, twice the lattice spacing of the microscopic model. This discretization ties the number of layers to the linear size: $N = L/2a$. (Equivalently $N = L/2$ given we set $a = 1$). The gauge fields $A_a$ need to be rescaled properly

$$A_a(t, x_{13}^{\sigma_1}, x_{23}^{\sigma_2}, x_\perp) \to \frac{1}{\sqrt{2a}} A_a^i(t, x_{13}^{\sigma_1}, x_{23}^{\sigma_2}) \,. \tag{58}$$

The action (54) becomes

$$S = \sum_{i=1}^{N} \int dt\, dx_{13}^{\sigma_1}\, dx_{23}^{\sigma_2}\, \frac{k}{\pi} A_1^i\, \partial_0 A_2^i + \cdots \,. \tag{59}$$

Thus we end up with $N$ copies of (2+1)-dimensional theories.

The dimension of the gauge fields in mass units is $[A_a] = D/2$. After discretization, the rescaled fields in (58) have dimension $[A_a^i] = \frac{D-1}{2}$. In particular, $[A_a^i] = 1$ for $D = 3$. Therefore, for each of the layers, we can define the holonomies

$$\exp\left(i \int_0^{l_i} dx_{13}^{\sigma_1} A_1^i\right) \quad \text{and} \quad \exp\left(i \int_0^{l_i} dx_{23}^{\sigma_2} A_2^i\right), \tag{60}$$

where $l_i$ is the size of each cycle of the 2-torus, and the arguments of the exponentials are properly dimensionless. These objects are gauge-invariant. In fact, under a gauge transformation, the fields transform as

$$A_1^i \to A_1^i + \partial_{13}^+ \partial_{13}^- \zeta^i \quad \text{and} \quad A_2^i \to A_2^i + \partial_{23}^+ \partial_{23}^- \zeta^i \,. \tag{61}$$

Let us analyse, say, the first holonomy in (60). Under a gauge transformation, it changes by a factor

$$\exp\left(i \int_0^{l_i} dx_{13}^{\sigma_1} \partial_{13}^+ \partial_{13}^- \zeta^i\right) = \exp\left(i \partial_{13}^{-\sigma_1} \zeta^i \Big|_{x_{13}^{\sigma_1}=0}^{x_{13}^{\sigma_1}=l_i}\right) \equiv 1 \,. \tag{62}$$

The above condition is satisfied with the general periodic boundary condition

$$\zeta^i \Big|_{x_{13}^{\sigma_1}=l_i} - \zeta^i \Big|_{x_{13}^{\sigma_1}=0} = 2\pi n_1^i x_{13}^{-\sigma_1}, \quad n_1^i \in \mathbb{Z} \,. \tag{63}$$

Infinitesimal gauge transformations correspond to $n_1^i = 0$, whereas $n_1^i \neq 0$ are associated with large gauge transformations. Similarly, for the second holonomy in (60), we obtain

$$\zeta^i \Big|_{x_{23}^{\sigma_2}=l_i} - \zeta^i \Big|_{x_{23}^{\sigma_2}=0} = 2\pi n_2^i x_{23}^{-\sigma_2}, \quad n_2^i \in \mathbb{Z} \,. \tag{64}$$

A large gauge transformation satisfying all these conditions can be constructed explicitly,

$$\zeta^i = \frac{2\pi n_1^i}{l_i} x_{13}^+ x_{13}^- + \frac{2\pi n_2^i}{l_i} x_{23}^+ x_{23}^-, \quad n_1^i, n_2^i \in \mathbb{Z} \,. \tag{65}$$

This implies an equivalence for the gauge fields

$$A_1^i \cong A_1^i + \frac{2\pi}{l_i} m_1^i \quad \text{and} \quad A_2^i \cong A_2^i + \frac{2\pi}{l_i} m_2^i, \quad m_1^i, m_2^i \in \mathbb{Z}. \tag{66}$$

Now we consider the ground state configuration, which corresponds to solutions depending only on the time,

$$A_a^i(t, x_{13}^+, x_{23}^-) = \frac{1}{l_i} \bar{A}_a^i(t). \tag{67}$$

Plugging this equation into the action (59) we obtain

$$S = \sum_{i=1}^N \int dt \, \frac{k}{\pi} \bar{A}_1^i \, \partial_0 \bar{A}_2^i. \tag{68}$$

The holonomies become

$$e^{i\bar{A}_1^i} \quad \text{and} \quad e^{i\bar{A}_2^i}. \tag{69}$$

From the action (67) it follows the commutation rule

$$[\bar{A}_1^i, \bar{A}_2^j] = -\frac{i\pi}{k} \delta^{ij}, \tag{70}$$

leading to the commutation relation between the holonomies

$$e^{i\bar{A}_1^i} e^{i\bar{A}_2^i} = e^{i\bar{A}_2^i} e^{i\bar{A}_1^i} e^{\frac{i\pi}{k}}, \tag{71}$$

which implies a $2k$-fold degeneracy for each plane $i$. The degeneracy of the layered system is then

$$(2k)^N. \tag{72}$$

Finally, taking into account that we have 4 sub-dimensional manifolds where charge is conserved, the total degeneracy is

$$\text{GSD} = (2k)^{4N}. \tag{73}$$

Using that $N = L/2$, we recover the degeneracy of the lattice model in $D = 3$. For $k = 1$ it agrees with the result of [2]: $2^{2L}$.

Now let us discuss how this generalizes to higher dimensional spaces. For concreteness, we consider the (5+1)-dimensional action

$$S = \int dt \, d^5x \, \frac{1}{2\pi} K_{ab} A_a \, \partial_0 A_b + \cdots. \tag{74}$$

In this case, charge is conserved in the following $2^4$ sub-dimensional manifolds with the corresponding measures,

$$\int dx_{15}^{\sigma_1} \, dx_{25}^{\sigma_2} \, dx_{35}^{\sigma_3} \, dx_{45}^{\sigma_4}. \tag{75}$$

We proceed similarly to the previous case, i.e., we write the action in terms of the coordinates of a sub-manifold where charge is conserved plus a perpendicular direction $x_\perp$, which is then discretized. With this, the action (74) becomes

$$S = \sum_{i=1}^N \int dt \, dx_{15}^{\sigma_1} \, dx_{25}^{\sigma_2} \, dx_{35}^{\sigma_3} \, dx_{45}^{\sigma_4} \, \frac{1}{2\pi} K_{ab} A_a^i \, \partial_0 A_b^i + \cdots, \tag{76}$$

where the fields $A_a^i$ were rescaled as in (58).

Now, the key point is that we can rotate the matrix $K$ according to (17) to bring it to the block-diagonal form

$$QKQ^T = \text{Diag}\left\{\begin{pmatrix} 0 & k_1 \\ -k_1 & 0 \end{pmatrix}, \begin{pmatrix} 0 & k_2 \\ -k_2 & 0 \end{pmatrix}\right\}, \tag{77}$$

where $k_1$ and $k_2$ are real and positive. In this basis, the fields $A_a^i$ decouple pairwise,

$$S = \sum_{i=1}^{N} \int dt\, dx_{15}^{\sigma_1}\, dx_{25}^{\sigma_2}\, dx_{35}^{\sigma_3}\, dx_{45}^{\sigma_4} \left[\frac{k_1}{\pi} A_1^i\, \partial_0 A_2^i + \frac{k_2}{\pi} A_3^i\, \partial_0 A_4^i + \cdots\right]. \tag{78}$$

Thus, we can construct the following pairs of holonomies

$$\exp\left(i\, l_i \int_0^{l_i} dx_{15}^{\sigma_1} A_1^i\right) \quad \text{and} \quad \exp\left(i\, l_i \int_0^{l_i} dx_{25}^{\sigma_2} A_2^i\right), \tag{79}$$

and

$$\exp\left(i\, l_i \int_0^{l_i} dx_{35}^{\sigma_3} A_3^i\right) \quad \text{and} \quad \exp\left(i\, l_i \int_0^{l_i} dx_{45}^{\sigma_4} A_4^i\right). \tag{80}$$

Notice that we have introduced an appropriate factor of $l_i$ in order to have a dimensionless argument in the exponentials[3]. The above holonomies correspond to the decomposition of the 4-dimensional torus $T^4$ in $T^4 = T^2 \times T^2$. Therefore, by proceeding in the same way as in the case $D = 3$, we see that these holonomies lead to a $(2k_1 \times 2k_2)$-fold degeneracy in each layer. For $N$ layers, we get

$$(2k_1 \times 2k_2)^N. \tag{81}$$

Finally, considering the $2^4$ sub-dimensional manifolds, it follows that the total ground state degeneracy is

$$\text{GSD} = (2k_1 \times 2k_2)^{2^4 N} = \left[2^2 \, \text{Pf}(K)\right]^{2^4 N}, \tag{82}$$

which is expressed in a basis-independent way in terms of the Pfaffian of the original matrix $K$.

The generalization to the odd $D$-dimensional case is immediate. We decompose the space in a $(D-1)$-dimensional sub-manifold corresponding to one of the $2^{D-1}$ sub-spaces where charge is conserved, and a perpendicular dimension which is then discretized. Next, we make the transformation (17) to bring the matrix $K$ to the block-diagonal form

$$QKQ^T = \text{Diag}\left\{\begin{pmatrix} 0 & k_1 \\ -k_1 & 0 \end{pmatrix}, \begin{pmatrix} 0 & k_2 \\ -k_2 & 0 \end{pmatrix}, \ldots, \begin{pmatrix} 0 & k_{\frac{D-1}{2}} \\ -k_{\frac{D-1}{2}} & 0 \end{pmatrix}\right\}, \tag{83}$$

where all $k$'s are real and positive. In this basis, the fields $A_a$ decouple pairwise, which is equivalent to decomposing the $(D-1)$-dimensional torus as

$$T^{D-1} = \underbrace{T^2 \times T^2 \times \cdots \times T^2}_{\frac{D-1}{2}}. \tag{84}$$

The corresponding degeneracy is

$$2k_1 \times 2k_2 \times \cdots \times 2k_{\frac{D-1}{2}} = 2^{\frac{D-1}{2}} \, \text{Pf}(K). \tag{85}$$

---

[3]In an arbitrary odd $D$-dimensional space, as $[A_a^1] = \frac{D-1}{2}$, we shall include the factor $l_i^{\frac{D-3}{2}}$ in order to make the argument dimensionless, i.e., the holonomies are of the form: $\exp\left(i\, l_i^{(D-3)/2} \int_0^{l_i} dx_{aD}^{\sigma_a} A_a^i\right)$, with $a = 1, 2, \ldots D-1$.

Taking into account the $N$ layers, we have

$$\left[2^{\frac{D-1}{2}} \operatorname{Pf}(K)\right]^N .\tag{86}$$

Finally, considering all the $2^{D-1}$ sub-dimensional manifolds, we obtain the total ground state degeneracy

$$\text{GSD} = \left[2^{\frac{D-1}{2}} \operatorname{Pf}(K)\right]^{2^{D-1}N} .\tag{87}$$

For the case of Clifford fractons, where $k_1 = k_2 = \cdots = k_{\frac{D-1}{2}} = 1$ or, equivalently, $\operatorname{Pf}(K) = 1$, the ground state degeneracy reduces to

$$\text{GSD} = 2^{(D-1)2^{D-3}L} ,\tag{88}$$

where we have again used that $N = L/2$. This is precisely the result shown in the end of Sec. 2 obtained directly from the lattice model.

## 5 Final Remarks

In this work we constructed fracton models on the lattice and identified their continuum description in terms of Chern-Simons-like theories. The construction is generic in that it applies to any system whose microscopic Hamiltonian is a sum of commuting projectors built from tensor products of spin-1/2 operators. Instead of working directly with tensor products of Pauli operators that represent the local variables, we utilize the Dirac representation of Clifford algebras. This representation makes a connection between the lattice model and the field theory simple. Our formalism can, in principle, be used to analyze other lattice models, such as those that exhibit subsystem symmetry protected topological (SSPT) phases [45,52] or type II fracton phases [3]. Applying this formalism to these problems is a natural direction for future work.

In the field theory, the algebraic structure of the Dirac matrices is encoded in an anti-symmetric matrix $K$. The details about an specific lattice model enter via this matrix $K$ (whose dimension depends on the size of the representation), the charge vectors $T$ (that specify the operators that are placed on the sites), as well as the lattice vector positions of the sites themselves. Given these data, one can follow the prescription here presented and derive an effective field theory for any type of Clifford-like fracton, such as the 3D Chamon (with a $2 \times 2$ Dirac representation) or the 3D Haah (with a $4 \times 4$ Dirac representation) codes. As a concrete example, we built fracton theories in odd $D$ spatial dimensional spaces. We discussed the properties of the resulting Chern-Simons-like theory, such as their currents, which are conserved in sub-manifolds, and the topological degeneracy of the ground states, which formally depends on the Pfaffian of the matrix $K$ and, as usual in fracton systems, on the linear size of the system.

Properties such as the mutual statistics of the quasiparticles where not explored in the present work. The restricted mobility of fractons makes it unnatural to speak of standard braiding. However, the authors in [53] were able to develop a theory of fusion and statistical processes that incorporates the mobility restrictions common in fracton models. An interesting question for future exploration is how our formalism could incorporate their notion of statistics.

For readers familiar with the $K$-matrices and charge vectors $T$ appearing in the description of Abelian fractional quantum Hall states [51], as well as their quantum wire constructions [54–57], it is tempting to expect that the description here presented – for "integer" fractons given our $K$ and $T$'s – could possibly lend itself to the analysis of fractional fractons. This is

an intriguing possibility that merits further investigation, but keeping the following points in mind.

The approach of this paper resembles quantum wire constructions of topological phases, but instead of wires we deploy $(0+1)$-dimensional degrees of freedom, i.e., ours is a "quantum dot" construction. Like in the wire constructions, we identify families of commuting operators that can be simultaneously pinned and gap the system. In the wire systems, fractionalization already takes place in the $(1+1)$-dimensional building blocks, and it is carried over to higher dimensions by coupling the wires, notably using only integer charge transfer operators. However, there is no fractionalization in the quantum dots of the construction of this paper. Of course, one may generalize the construction presented here to start with wires instead of dots, in which case fractionalization may appear more easily.

**Added note** : It has been brought to our attention that the word "fracton" has been used in physics in other contexts before. An early use was in [58] in reference to fractional charges in quantum chromodynamics. In our construction, we adopt the modern meaning of the word as stressed in the main text.

# Acknowledgements

This work is supported by the Brazilian agency Coordenação de Aperfeiçoamento de Pessoal de Nível Superior (CAPES) under grant number 88881.361635/2019-01 (W. F.), the CNPq grant number 311149/2017-0 (P. G.), and the DOE Grant No. DE-FG02-06ER46316 (C .C). W. F. acknowledges support by the Condensed Matter Theory Visitors program at Boston University.

# A   Euclidean Dirac matrix representations of Clifford algebras

We construct fracton models in odd $D = 2n + 1$ dimensions using representations of the Clifford algebra. Specifically, we use the Euclidean Dirac matrices. Below we construct these representations and show properties that these matrices satisfy. These properties are used, for example, to argue that there is no operator that can move defects in the corresponding fracton models.

Let us work with matrices defined as the tensor products of Pauli matrices:

$$\gamma^{(n)}_{\mu_1\mu_2\ldots\mu_n} \equiv \sigma_{\mu_1} \otimes \sigma_{\mu_2} \otimes \cdots \otimes \sigma_{\mu_n}, \tag{89}$$

with $\mu_i = 0, 1, 2, 3$ and $\sigma_0 \equiv \mathbb{1}$. We shall obtain a set of $2n + 1$ mutually anticommuting matrices for any $n$. We construct this set inductively.

For $n = 1$, the set contains the matrices $\gamma^{(1)}_1 = \sigma_1$, $\gamma^{(1)}_2 = \sigma_2$, and $\gamma^{(1)}_3 = \sigma_3$. Equivalently, we can label these matrices as $\gamma^{(1)}_I$, with indices $I \in S^{(1)} = \{1, 2, 3\}$.

For $n = 2$, we first construct the following 3 matrices using the $\gamma^{(1)}_I$, $I \in S^{(1)}$: $\gamma^{(2)}_{I3} = \gamma^{(1)}_I \otimes \sigma_3$. Second, we take the following two matrices: $\gamma^{(2)}_{01}$ and $\gamma^{(2)}_{02}$. Therefore the five matrices $\gamma^{(2)}_I$, with indices $I \in S^{(2)} = \{13, 23, 33, 01, 02\}$, are all anticommuting.

We proceed by induction. Suppose that we have $2n - 1$ anticommuting matrices $\gamma^{(n-1)}_i$, $i \in S^{(n-1)}$. First, using the $2n - 1$ matrices $\gamma^{(n-1)}_i$, $i \in S^{(n-1)}$, build the matrices

$$\gamma^{(n)}_{i3} = \gamma^{(n-1)}_i \otimes \sigma_3.$$

Second, take the two matrices

$$\gamma^{(n)}_{0\ldots01} \quad \text{and} \quad \gamma^{(n)}_{0\ldots02}.$$

The $2n - 1 + 2 = 2n + 1$ matrices $\gamma_I^{(n)}$, with $I \in S^{(n)} = \{i3 \mid i \in S^{(n-1)}\} \cup \{0 \dots 0\, 1, \, 0 \dots 0\, 2\}$ are all anticommuting.

These $2n + 1$ matrices multiply to the identity up to a prefactor:

$$\prod_{I \in S^{(n)}} \gamma_I^{(n)} = \pm i^n \; \gamma_{0 \dots 0 0}^{(n)} \,, \tag{90}$$

where the $\pm$ simply depends on the order that the matrices are multiplied (the choice of order of the indices $I \in S^{(n)}$). This relation can also be proved by induction. Notice that it holds for $n = 1$. If it holds for $n - 1$, then it follows that

$$\begin{aligned}
\prod_{I \in S^{(n)}} \gamma_I^{(n)} &= \left( \prod_{i \in S^{(n-1)}} \gamma_{i3}^{(n)} \right) \gamma_{0 \dots 0 1}^{(n)} \, \gamma_{0 \dots 0 2}^{(n)} \\
&= \left( \pm i^{n-1} \, \gamma_{0 \dots 0 3}^{(n)} \right) \gamma_{0 \dots 0 1}^{(n)} \, \gamma_{0 \dots 0 2}^{(n)} \\
&= \mp i^n \, \gamma_{0 \dots 0 0}^{(n)} \,.
\end{aligned} \tag{91}$$

This property means that the last, or $(2n + 1)$th, $\gamma$-matrix can be obtained from the product of all the other $2n$ matrices. It also follows that any matrix that is a tensor product of Pauli matrices can be written as products of these $2n$ $\gamma$-matrix. (Notice that there are $4^n$ possible tensor products of Pauli matrices, and $2^{2n} = 4^n$ choices of whether a $\gamma$-matrix enters or not the product of $\gamma$'s.)

The construction above yields a set of $2n + 1$ matrices $\gamma_I^{(n)}$ satisfying

$$\{\gamma_I^{(n)}, \gamma_J^{(n)}\} = 2 \, \delta_{IJ} \,. \tag{92}$$

The set of indices $I \in S^{(n)}$ can be interchanged to $I = 1, \dots, 2n + 1$, which is the notation we use in the main text for the Euclidean Dirac matrices.

## A.1 Properties of the Euclidean Dirac matrices

Let us now show three useful properties of the $2n + 1$ matrices $\gamma_I^{(n)}$ with $I \in S^{(n)}$.

1. **The identity is the only tensor product of Pauli matrices that commutes with all the Dirac matrices.** In other words, only the matrix $\gamma_J^{(n)}$, $J = 00 \dots 0$, can commute the $2n + 1$ matrices $\gamma_I^{(n)}$ with $I \in S^{(n)}$.

   To show this property, suppose that there is a matrix $\gamma_J^{(n)}$ that commutes with all the $2n + 1$ matrices. This $J$ must be of the form $J = j0$ for $\gamma_J^{(n)}$ to commute with both $\gamma_{0 \dots 0 1}^{(n)}$ and $\gamma_{0 \dots 0 2}^{(n)}$. Therefore,

   $$[\gamma_J^{(n)}, \gamma_I^{(n)}] = 0, \quad \forall I \in S^{(n)} \quad \Leftrightarrow \quad [\gamma_j^{(n-1)}, \gamma_i^{(n-1)}] = 0, \quad \forall i \in S^{(n-1)} \,.$$

   We can use this recursion all the way to $n = 1$, where only $\gamma_0^{(1)}$ commutes with the $\gamma_i^{(1)}, i \in S^{(1)}$, and conclude that $J$ must be $J = 00 \dots 0$, i.e., all the entries must be 0.

2. **The set of matrices $\gamma_I^{(n)}$ with $I \in S^{(n)}$ is maximal, i.e., no other matrix can be added to the set that anticommutes with those already in.** The statement is true for $n = 1$: the matrices $\gamma_I^{(1)}$ with $I \in S^{(1)}$ are the three Pauli matrices, leaving no other option to include that would anticommute with these three.

Now suppose that the statement is true up to $n-1$; let us analyze the consequences for when we consider $n$.

Suppose by contradiction that there exists a $J \notin S^{(n)}$ such that $\gamma_J^{(n)}$ anticommutes with all the $\gamma_I^{(n)}$ with $I \in S^{(n)}$. Let us break the problem in four cases, and show impossibility in all cases.

- $J = j0$

  This is the simplest case: $\gamma_{j0}^{(1)}$ commutes with both $\gamma_{0\ldots01}^{(n)}$ and $\gamma_{0\ldots02}^{(n)}$, so $J = j0$ cannot be added to the set.

- $J = j1$ (the case $J = j2$ is analogous)

  This case is also simple: $\gamma_{j1}^{(1)}$ commutes with $\gamma_{0\ldots01}^{(n)}$, so $J = j1$ cannot be added to the set.

- $J = j3$

  $\gamma_{j3}^{(1)}$ anticommutes with $\gamma_{0\ldots01}^{(n)}$ and $\gamma_{0\ldots02}^{(n)}$, so we should only consider the anticommutation with the other $2n-1$ matrices $\gamma_{i3}^{(n)}$, for $i \in S^{(n-1)}$. But because $S^{(n-1)}$ is maximal, there is no new $j \notin S^{(n-1)}$ to add.

We thus conclude that the set of $2n+1$ matrices $\gamma_I^{(n)}$ with $I \in S^{(n)}$ is maximal.

3. **There is no matrix $\gamma_J^{(n)}$ that commutes with $2n$ of the matrices $\gamma_I^{(n)}$ with $I \in S^{(n)}$. Therefore, defects cannot be created in only one direction**. Phrasing it differently, this states that there is no line defects on the model. This result will allow us to argue that we can construct a fracton model.

   The statement is true for $n = 1$: there is no matrix $\gamma_J^{(1)}$ that commutes with two of the matrices $\gamma_I^{(1)}$ with $I \in S^{(1)}$, because no one Pauli matrix commutes with two Pauli matrices.

   Now suppose that the statement is true up to $n-1$; let us analyze the consequences for when we consider $n$.

   Let us break the problem in four cases:

   - $J = j0$

     In this case, the commutation with $\gamma_{0\ldots01}^{(n)}$ and $\gamma_{0\ldots02}^{(n)}$ comes for free. Therefore we reduce the problem to finding $\gamma_j^{(n-1)}$ that commutes with $2(n-1)$ matrices $\gamma_i^{(n-1)}$ with $i \in S^{(n-1)}$. Since there is no solution for this problem (the statement is true for the case with $n-1$), then there is no solution for the case with $n$ either.

   - $J = j3$

     This is the simplest case; $\gamma_{j3}^{(1)}$ anticommutes with $\gamma_{0\ldots01}^{(n)}$ and $\gamma_{0\ldots02}^{(n)}$, so it is impossible that there are $2n$ other matrices that commute with $\gamma_J^{(n)}$ among the $\gamma_I^{(n)}$ with $I \in S^{(n)}$, since there are at most $2n+1-2 = 2n-1 < 2n$.

   - $J = j1$ (the case $J = j2$ is analogous)

     $\gamma_{j1}^{(1)}$ commutes with $\gamma_{0\ldots01}^{(n)}$ and anticommutes with $\gamma_{0\ldots02}^{(n)}$. So we need to find $2n-1$ additional matrices that commute with $\gamma_{j1}^{(n)}$ among the $\gamma_{i3}^{(n)}$ with $i \in S^{(n-1)}$. This is equivalent to finding $2n-1$ matrices that anticommute with $\gamma_j^{(n-1)}$ among the $\gamma_i^{(n-1)}$ with $i \in S^{(n-1)}$. This is impossible since the set $S^{(n-1)}$ is maximal (see above).

# B   Fracton models build from the Clifford algebra representations

We can construct a fracton model in $D$ dimensions if $D$ is odd. In this case we take $n = (D-1)/2$ and we can use the matrices $\gamma_I^{(n)}$, $I \in S^{(n)}$ in the construction. Here we shall label these $2n+1$ matrices simply $\gamma^I$, $I = 1, \ldots, 2n+1$, as we did in the main text.

The construction can be made in the $D$-dimensional hypercube, with orthogonal basis vectors $\hat{a}_I$, $I = 1, \ldots, D$, as presented in the main text. We place the degrees of freedom on the even sublattice $\Lambda_e$. The dimension of the local Hilbert space at each site is $2^n$, or equivalently, that associated with the $n$ spins or gradings of Pauli operators used to construct the $\gamma$-matrix representations. At these even sublattice sites we place operators $\Gamma^{(I,\alpha)}$, $\alpha = 1, \ldots, n$ built as products of the $\gamma$-matrices (in turn built from tensor products of Pauli matrices).

The first set of operators, with $\alpha = 1$, is the set of Dirac matrices $\gamma^I$, $I = 1, \ldots, 2n+1$, or explicitly

$$\Gamma^{(I,1)} = \gamma^I . \tag{93}$$

The other sets are needed to gap the model.

We define $\mathcal{O}_{\vec{x}}^{(\alpha)}$ operators centered at sites $\vec{x}$ on the odd sublattice $\Lambda_o$,

$$\mathcal{O}_{\vec{x}}^{(\alpha)} \equiv \prod_{I=1}^{D} \Gamma_{\vec{x}-\hat{a}_I}^{(I,\alpha)} \, \Gamma_{\vec{x}+\hat{a}_I}^{(I,\alpha)} , \quad \alpha = 1, \ldots, \frac{(D-1)}{2} , \tag{94}$$

and using these the Hamiltonian

$$H = -\sum_{\alpha=1} \left( g_\alpha \sum_{\vec{x}} \mathcal{O}_{\vec{x}}^{(\alpha)} \right) . \tag{95}$$

We can choose the operators $\Gamma^{(I,\alpha)}$ such that

$$\left[ \mathcal{O}_{\vec{x}}^{(\alpha)} , \mathcal{O}_{\vec{x}'}^{(\beta)} \right] = 0 , \quad \forall \alpha, \beta , \forall \vec{x}, \vec{x}' . \tag{96}$$

As stated in the main text, in this case *i*) the Hamiltonian is a sum of commuting projectors, and *ii*) there are as many commuting projectors as the number of degrees of freedom in the problem (up to constraints tied to the topological degeneracy).

Let us first focus on the operators $\mathcal{O}^{(1)}$ for simplicity. These are defined as

$$\mathcal{O}_{\vec{x}}^{(1)} = \prod_{I=1}^{D} \gamma_{\vec{x}-\hat{a}_I}^{I} \, \gamma_{\vec{x}+\hat{a}_I}^{I} . \tag{97}$$

The operators $\gamma_{\vec{x}}^I$ satisfy the following commutation relations:

$$\{\gamma_{\vec{x}}^I , \gamma_{\vec{x}'}^J\} = 2\,\delta_{IJ} , \text{ if } \vec{x} = \vec{x}' , \qquad \text{and} \qquad [\gamma_{\vec{x}}^I , \gamma_{\vec{x}'}^J] = 0 , \text{ if } \vec{x} \neq \vec{x}' . \tag{98}$$

(We remark that these models are bosonic, and not fermionic; the Dirac matrices represent the tensor product of local Pauli matrices, that in turn represent spin degrees of freedom on the lattice.) Given these commutation relations, it follows that all distinct $\mathcal{O}_{\vec{x}}^{(1)}$ and $\mathcal{O}_{\vec{x}'}^{(1)}$ that share common sites commute: 1) they either share a single site along the line that connects them, in which case the same operator (same $I$), or 2) they share two sites with different components $I$ and $J$ entering in each of $\mathcal{O}_{\vec{x}}^{(1)}$ and $\mathcal{O}_{\vec{x}'}^{(1)}$, and hence there is a factor of $-1$ from the anti-commutation relation of each common site, and hence in total a factor $(-1)^2$, leading to the commutation of the two operators.

The operators $\mathcal{O}_{\vec{x}}^{(1)}$ square to unity, and thus have eigenvalues $\pm 1$. The ground state has all eigenvalues $+1$ for all operators. Excitations correspond to eigenvalues $-1$. Because we used all the $2n+1$ Dirac matrices in constructing $\mathcal{O}_{\vec{x}}^{(1)}$, and as demonstrated above in Sec. A.1, there is *no* operator that anti-commutes with one and only one of the $\gamma^I$. Therefore, it is not possible to construct a local operator whose sole effect is to create a pair of defects, or move a single defect. Defects are only created in at least quadruplets in any dimension $D = 2n+1$, much as in the $D = 3$ model in Ref. [1]. This property that defects cannot be created in pairs, but only in at least quadruplets, underscores the fracton nature of these odd $D$ models.

Let us now discuss the other operators $\mathcal{O}_{\vec{x}}^{(\alpha)}$, $\alpha = 2, \ldots, n$. The argument for the commutativity follows a similar line. When two operators $\mathcal{O}_{\vec{x}}^{(\alpha)}$ and $\mathcal{O}_{\vec{x}'}^{(\beta)}$ share sites, there are two cases to consider.

The case when they share one site: the neighboring $\mathcal{O}$'s, defined at sites $\vec{x}$ and $\vec{x} + 2\hat{a}_I$ of $\Lambda_o$, share the $\Lambda_e$ site at $\vec{x} + \hat{a}_I$, and they commute if

$$\left[ \Gamma^{(I,\alpha)}, \Gamma^{(I,\beta)} \right] = 0. \tag{99}$$

The case when they share two sites: the neighboring $\mathcal{O}^{(\alpha)}$'s, defined at sites $\vec{x}$ and $\vec{x} + \hat{a}_I + \hat{a}_J$ of $\Lambda_o$, share the two $\Lambda_e$ sites at $\vec{x} + \hat{a}_I$ and $\vec{x} + \hat{a}_J$. The operators on those sites either commute or anti-commute, which can be cast as

$$\Gamma^{(I,\alpha)} \Gamma^{(J,\beta)} = (-1)^{\eta_{IJ}^{(\alpha\beta)}} \Gamma^{(J,\beta)} \Gamma^{(I,\alpha)}, \tag{100}$$

with $\eta_{IJ}^{(\alpha\beta)} = 0$ or $1$, and the desired commutation relations Eq. (96) are guaranteed if

$$\eta_{IJ}^{(\alpha\beta)} = \eta_{JI}^{(\alpha\beta)}. \tag{101}$$

This condition, or equivalently that $\eta_{IJ}^{(\alpha\beta)} + \eta_{JI}^{(\alpha\beta)} = 0 \mod 2$, is the counterpart to $C_{IJ}^{(\alpha\beta)} = 0$ of Eq. (28) in the main text. The components $T^{(I,\alpha)}$ of the $T$-vectors, when computed mod 2, simply encode which of the $\gamma^I$ matrices enter in the product defining the operator $\Gamma^{(I,\alpha)}$. Because of this relation, we shall show how to construct the $\Gamma^{(I,\alpha)}$'s by showing how to ensure $C_{IJ}^{(\alpha\beta)} = 0$, which we can solve more easily using integer instead of binary vectors.

Explicitly, we construct the operators $\Gamma^{(I,\alpha)}$ using $2n$-dimensional $T$-vectors, $T_a^{(I,\alpha)}, a = 1, \ldots, 2n$, as follows:

$$\Gamma^{(I,\alpha)} = \left( \gamma^1 \right)^{T_1^{(I,\alpha)}} \left( \gamma^2 \right)^{T_2^{(I,\alpha)}} \cdots \left( \gamma^{2n} \right)^{T_{2n}^{(I,\alpha)}}. \tag{102}$$

Notice that since $(\gamma^I)^2 = 1$, only the values of the $T$-vectors mod 2 matter. The particular case of the first set, see Eq. (103), corresponds to the vector

$$T_a^{(I,1)} = t_a^I = \delta_a^I, \ I = 1, \ldots, 2n, \quad \text{and} \quad T_a^{(2n+1,1)} = -\sum_{I=1}^{2n} t_a^I. \tag{103}$$

(The $t^I$ are the basis vectors.)

The commutation relations between the $\gamma$-matrices can be encoded in an integer-valued anti-symmetric $K$-matrix via

$$\gamma^I \gamma^J = e^{i\pi t_a^{(I)} K_{ab} t_b^{(J)}} \gamma^J \gamma^I, \tag{104}$$

where repeated index summation over the $a$ and $b$ are used. The correct commutation relations follow from requiring that

$$t_a^{(I)} K_{ab} t_b^{(J)} = \begin{cases} 0 \mod 2, & I = J \\ 1 \mod 2, & I \neq J \end{cases}. \tag{105}$$

It follows that the commutation relations

$$\Gamma^{(I,\alpha)} \, \Gamma^{(J,\beta)} = e^{i\pi \, T_a^{(I,\alpha)} K_{ab} \, T_b^{(J,\beta)}} \, \Gamma^{(J,\beta)} \, \Gamma^{(I,\alpha)} \,, \tag{106}$$

or equivalently, using Eq. (100),

$$\eta_{IJ}^{(\alpha\beta)} = T_a^{(I,\alpha)} K_{ab} \, T_b^{(J,\beta)} \mod 2 \,. \tag{107}$$

Then, condition Eq. (101) is equivalent to

$$C_{IJ}^{(\alpha\beta)} = 0 \mod 2 \,, \tag{108}$$

where

$$C_{IJ}^{(\alpha\beta)} \equiv T_a^{(I,\alpha)} K_{ab} \, T_b^{(J,\beta)} + T_a^{(J,\alpha)} K_{ab} \, T_b^{(I,\beta)} \,. \tag{109}$$

While we just need $C_{IJ}^{(\alpha\beta)}$ to vanish mod 2, we can simply demand that it vanishes, and still solve the problem as we show below.

Let us now construct vectors $T_a^{(I,\alpha)}$ that satisfy $C_{IJ}^{(\alpha\beta)} = 0$. We already have the first set of $T$-vectors from Eq. (103). Now build the other sets of $T$-vectors via a family of linear transformations $L^{(\alpha)}$:

$$T_a^{(I,\alpha)} = \sum_M L_{IM}^{(\alpha)} T_a^{(M,1)} = L_{Ia}^{(\alpha)} \,, \quad I, M = 1, \ldots, 2n \,, \quad \text{and} \quad T_a^{(2n+1,\alpha)} = -\sum_{I=1}^{2n} T_a^{(I,\alpha)} \,. \tag{110}$$

It follows, for $I, J = 1, \ldots, 2n$, that

$$\begin{aligned}
C_{IJ}^{(\alpha\beta)} &= T_a^{(I,\alpha)} K_{ab} \, T_b^{(J,\beta)} + T_a^{(J,\alpha)} K_{ab} \, T_b^{(I,\beta)} \\
&= L_{Ia}^{(\alpha)} K_{ab} \, L_{Jb}^{(\beta)} + L_{Ja}^{(\alpha)} K_{ab} \, L_{Ib}^{(\beta)} \\
&= (L^{(\alpha)} \, K \, L^{(\beta)^\top})_{IJ} + (L^{(\beta)} \, K^\top \, L^{(\alpha)^\top})_{IJ} \,,
\end{aligned} \tag{111}$$

or equivalently, that

$$\begin{aligned}
C^{(\alpha\beta)} &= L^{(\alpha)} \, K \, L^{(\beta)^\top} + L^{(\beta)} \, K^\top \, L^{(\alpha)^\top} \\
&= L^{(\alpha)} \, K \, L^{(\beta)^\top} + (L^{(\alpha)} \, K \, L^{(\beta)^\top})^\top \,.
\end{aligned} \tag{112}$$

Hence the condition that the commutation relations $C^{(\alpha\beta)}$ vanish require that the sets of $(\alpha, \beta)$-indexed matrices $(L^{(\alpha)} \, K \, L^{(\beta)^\top})$ be anti-symmetric (in the indices $I$ and $J$). Let then

$$L^{(\alpha)} \, K \, L^{(\beta)^\top} = A^{(\alpha\beta)} \,, \tag{113}$$

where the $A^{(\alpha\beta)}$ are anti-symmetric matrices for any of the $\alpha, \beta$ pairs. For given choices of matrices $A^{(\alpha\beta)}$, we can solve sequentially for

$$L^{(\alpha)} = A^{(\alpha\beta)} (L^{(\beta)^\top})^{-1} K^{-1} \,, \tag{114}$$

i.e., start with $\beta = 1$ and $L^{(1)} = \mathbb{1}$, obtain $L^{(2)}$ for some arbitrary choice of $A^{(21)}$, then for some choice $A^{(31)}$ obtain $L^{(3)}$, and so on. In other words, we can determine the $L^{(\alpha)}$ from using $\beta = 1$ and $L^{(1)} = \mathbb{1}$ in Eq. (114):

$$L^{(\alpha)} = A^{(\alpha 1)} K^{-1} \,, \tag{115}$$

for anti-symmetric choices of $A^{(\alpha 1)}$. Notice that the $L^{(\alpha)}$ cannot be equal, otherwise two sets of $T^{(I,\alpha)}$'s would be identical. The number of solutions (number of $\alpha$'s) depend on the dimension $D-1$ of the matrices, for example the matrix $K$. Notice that if $K$ is $2 \times 2$, any anti-symmetric matrix is proportional to $i\sigma_2$, and therefore it follows from Eq. (115) that one cannot get a non-trivial solution other than $L^{(1)} \propto \mathbb{1}$.

There are compatibility conditions for the matrices, because one can reach, for example, $L^{(3)}$ from $L^{(1)}$ or $L^{(2)}$. For example,

$$
\begin{aligned}
L^{(\alpha)} K L^{(\beta)\top} &= A^{(\alpha 1)} K^{-1} K K^{-1\top} A^{(\beta 1)\top} \\
&= A^{(\alpha 1)} K^{-1} A^{(\beta 1)},
\end{aligned} \tag{116}
$$

or equivalently

$$
A^{(\alpha\beta)} = A^{(\alpha 1)} K^{-1} A^{(\beta 1)}. \tag{117}
$$

### B.1 Example of $D = 5$

Consider the following $4 \times 4$ $K$-matrix:

$$
K_4 = \begin{bmatrix} 0 & +1 & +1 & +1 \\ -1 & 0 & +1 & +1 \\ -1 & -1 & 0 & +1 \\ -1 & -1 & -1 & 0 \end{bmatrix}, \tag{118}
$$

with inverse

$$
K_4^{-1} = \begin{bmatrix} 0 & -1 & +1 & -1 \\ +1 & 0 & -1 & +1 \\ -1 & +1 & 0 & -1 \\ +1 & -1 & +1 & 0 \end{bmatrix}. \tag{119}
$$

The choice $A^{(11)} = K_4$ yields $L^{(1)} = \mathbb{1}$, as it should be. Choose the anti-symmetric matrix $A^{(21)} = K_2 \otimes \mathbb{1}_2$, where $K_2 = \begin{pmatrix} 0 & +1 \\ -1 & 0 \end{pmatrix}$, or explicitly,

$$
A^{(21)} = \begin{bmatrix} 0 & 0 & +1 & 0 \\ 0 & 0 & 0 & +1 \\ -1 & 0 & 0 & 0 \\ 0 & -1 & 0 & 0 \end{bmatrix}, \tag{120}
$$

from which we obtain

$$
L^{(2)} = \begin{bmatrix} -1 & +1 & 0 & -1 \\ +1 & -1 & +1 & 0 \\ 0 & +1 & -1 & +1 \\ -1 & 0 & +1 & -1 \end{bmatrix}. \tag{121}
$$

From the $L$ matrix we obtain the vectors

$$
\begin{aligned}
T^{(1,2)} &= (-1,+1,\ 0,-1) \\
T^{(2,2)} &= (+1,-1,+1,\ 0) \\
T^{(3,2)} &= (\ 0,+1,-1,+1) \\
T^{(4,2)} &= (-1,\ 0,+1,-1) \\
T^{(5,2)} &= (+1,-1,-1,+1)\,.
\end{aligned}
\tag{122}
$$

The corresponding operators $\Gamma^{(I,2)}$ are:

$$
\begin{aligned}
\Gamma^{(1,2)} &= \gamma^1\,\gamma^2\,\gamma^4 \ \sim\ \gamma^3\,\gamma^5 \\
\Gamma^{(2,2)} &= \gamma^1\,\gamma^2\,\gamma^3 \ \sim\ \gamma^4\,\gamma^5 \\
\Gamma^{(3,2)} &= \gamma^2\,\gamma^3\,\gamma^4 \ \sim\ \gamma^1\,\gamma^5 \\
\Gamma^{(4,2)} &= \gamma^1\,\gamma^3\,\gamma^4 \ \sim\ \gamma^2\,\gamma^5 \\
\Gamma^{(5,2)} &= \gamma^1\,\gamma^2\,\gamma^3\,\gamma^4 \ \sim\ \gamma^5\,.
\end{aligned}
\tag{123}
$$

One can summarize the operators $\Gamma^{(I,\alpha)}$ in the following table, as we did in the main text:

| $\Gamma^{(I,\alpha)}$ | $I =$ | 1 | 2 | 3 | 4 | 5 |
|---|---|---|---|---|---|---|
| $\alpha = 1$ | | $\gamma^1$ | $\gamma^2$ | $\gamma^3$ | $\gamma^4$ | $\gamma^5$ |
| $\alpha = 2$ | | $\gamma^3\gamma^5$ | $\gamma^4\gamma^5$ | $\gamma^1\gamma^5$ | $\gamma^2\gamma^5$ | $\gamma^5$ |

$$\tag{124}$$

## B.2 Construction for general $D = 2n + 1$

Define the following $n \times n$ anti-symmetric matrix:

$$
K_n =
\begin{bmatrix}
0 & +1 & +1 & \dots & +1 \\
-1 & 0 & +1 & \dots & +1 \\
-1 & -1 & 0 & \dots & +1 \\
\vdots & \vdots & & \ddots & \vdots \\
-1 & -1 & \dots & -1 & 0
\end{bmatrix}_{n\times n}
\,.
\tag{125}
$$

The $2n \times 2n$ $K$-matrix we need for $D = 2n + 1$ is then simply $K_{2n}$.

The following anti-symmetric matrices $A^{(\alpha 1)}$

$$
\begin{aligned}
A^{(11)} &= K_{2n}\,, \\
A^{(\alpha 1)} &= (K_n)^{2\alpha-3} \otimes \mathbb{1}_2\,, \quad \alpha = 2,\dots,n\,,
\end{aligned}
\tag{126}
$$

commute with both $K_{2n}$ and $K_{2n}^{-1}$; using this property and the anti-symmetry of both the $A^{(\alpha 1)}$ and the $K_{2n}$, one can show that the $A^{(\alpha\beta)}$ obtained through Eq. (117) are anti-symmetric, as required.

With these $A^{(\alpha 1)}$, one can proceed to find the $L^{(\alpha)}$ matrices and then the vectors $T^{(I,\alpha)}$, and finally the operators $\Gamma^{(I,\alpha)}$ with the desired commutation relations.

## C  Degeneracy

In the main text we argued that the topological degeneracy of the model is at least $2^{(D-1)\,2^{D-2}}$. This number follows from the constraints of multiplying all the $\mathcal{O}^{(\alpha)}$ operators:

$$\prod_{\vec{x}\in\Lambda_{o,k}} \mathcal{O}^{(\alpha)}_{\vec{x}} = \mathbb{1}\,, \quad k=1,\dots,2^{D-1}, \quad \alpha=1,\dots,(D-1)/2\,, \tag{127}$$

where $k$ labels the $2^{D-1}$ sub-lattices $\Lambda_{o,k}$. (A unit cell of the hypercubic lattice contains $2^D$ sites, half of them are on the even and half on the odd sub-lattice – hence there are $2^{D-1}$ distinct sub-lattices of the odd sub-lattice.)

The degeneracy can be greater, and can depend on the system size. Here we follow Bravyi, Leemhuis, and Terhal's calculation in their appendix A of [2].

It follows from $\sum_{I=1}^{D} T_a^{(I,\alpha)} = 0$ that the $\Gamma^{(I,\alpha)}$ multiply to the identity (up to a factor of magnitude 1, that also depends on the order of multiplication). Using this property, we arrive at the equivalent of their parity checks:

$$\bigoplus_{\substack{J\neq I\\ q=\pm 1}} t^{(\alpha)}(\vec{x}+q\,\hat{a}_J) = 0\,, \quad \vec{x}\in\Lambda_e,\ I=1,2,\dots D\,, \quad \alpha=1,\dots,\frac{D-1}{2}\,. \tag{128}$$

(The $t$ above conforms to their notation; they are not related to our $t$-vectors.) Notice that, for each $\alpha$, the $D$ equations are linearly dependent, and that the sum over all of their left hand side is identically zero. Suming any pair of these equations yield

$$t^{(\alpha)}(\vec{x}-\hat{a}_I)\oplus t^{(\alpha)}(\vec{x}+\hat{a}_I)\oplus t^{(\alpha)}(\vec{x}-\hat{a}_J)\oplus t^{(\alpha)}(\vec{x}+\hat{a}_J)=0\,, \quad I,J=1,2,\dots,D\,. \tag{129}$$

The solutions of these equations for the case when $L_1=L_2=\cdots=L_D$ in a similar way as in 3-dimensions: first use two lines (with $2L/2=L$ sites on a sublattice) and generate the solution on a plane, then two planes and generate the solution in the 3rd dimension, and after that proceed accordingly, use 2 3-dimensional hyperplanes to generate the solutions in 4-dimensions, and so on. The number of logical quibts generated in this way is $\mathcal{C}_D = L/2 \times 2 \times 2 \times \cdots \times 2$, with $D-1$ 2's, i.e., $\mathcal{C}_D = 2^{D-2}L$. When we take into account all the $\alpha=1,\dots,(D-1)/2$, we have $(D-1)\,2^{D-3}\,L$ logical qubits. Therefore, the ground state degeneracy is

$$\mathrm{GSD} = 2^{(D-1)\,2^{D-3}\,L}\,. \tag{130}$$

## D  Level quantization from the effective field theory

Here we give the details of the calculations in 4.1. To understand quantization of the matrix $K$ for arbitrary dimensions, it is convenient to consider large gauge transformations depending only on time. In other words, we impose the condition $t\in[0,\tau)$, and place the system in a spatially closed manifold $\mathcal{M}^D$:

$$M = \mathcal{S}^1 \times \mathcal{M}^D\,. \tag{131}$$

As the gauge theory is compact, this implies flux quantization. According to the normalization used in the manuscript, this reads

$$\int_{\mathcal{M}^D} B^{(\alpha)}_{a_1 a_2 \dots a_{D-3}} \equiv \pi p^{(\alpha)}_{a_1 a_2 \dots a_{D-3}}, \quad p^{(\alpha)}_{a_1 a_2 \dots a_{D-3}} \in \mathbb{Z}\,, \tag{132}$$

where $B^{(\alpha)}_{a_1 a_2 \dots a_{D-3}}$ is the magnetic field given in (42), which we repeat here for convenience

$$B^{(\alpha)}_{a_1 a_2 \dots a_{D-3}} = \epsilon_{a_1 a_2 \dots a_{D-1}} \mathcal{D}^{(\alpha)}_{a_{D-2}} A_{a_{D-1}}\,. \tag{133}$$

This implies that $p^{(\alpha)}_{a_1 a_2 \ldots a_{D-3}}$ are completely anti-symmetric. It is useful to invert relation (133)

$$\mathcal{D}^{(\alpha)}_{b_1} A_{b_2} - \mathcal{D}^{(\alpha)}_{b_2} A_{b_1} = \frac{1}{(D-3)!} \epsilon_{a_1 a_2 \ldots a_{D-3} b_1 b_2} B^{(\alpha)}_{a_1 a_2 \ldots a_{D-3}} . \tag{134}$$

Now, let us consider large gauge transformations that wind around the $\mathcal{S}^1$ (time direction)

$$\zeta^{(\alpha)} \equiv 2\pi n^{(\alpha)} \frac{t}{\tau}, \quad n^{(\alpha)} \in \mathbb{Z} . \tag{135}$$

This implies

$$A^{(\alpha)}_0 \to A^{(\alpha)}_0 + 2\pi n^{(\alpha)} \frac{1}{\tau} . \tag{136}$$

The corresponding variation of the action is

$$
\begin{aligned}
\delta S &= \int d^D x \, dt \, \frac{1}{\pi} \sum_\alpha 2\pi n^{(\alpha)} \frac{1}{\tau} K_{ab} \mathcal{D}^{(\alpha)}_a A_b \\
&= \int d^D x \, dt \sum_\alpha n^{(\alpha)} \frac{1}{\tau} K_{ab} \left( \mathcal{D}^{(\alpha)}_a A_b - \mathcal{D}^{(\alpha)}_b A_a \right) \\
&= \int \frac{dt}{\tau} \sum_\alpha n^{(\alpha)} K_{ab} \frac{1}{(D-3)!} \epsilon_{a_1 a_2 \ldots a_{D-3} ab} \int d^D x \, B^{(\alpha)}_{a_1 a_2 \ldots a_{D-3}} .
\end{aligned}
\tag{137}
$$

By using the flux quantization (132), we obtain

$$\delta S = \pi \sum_\alpha n^{(\alpha)} K_{ab} \frac{1}{(D-3)!} \epsilon_{a_1 a_2 \ldots a_{D-3} ab} \, p^{(\alpha)}_{a_1 a_2 \ldots a_{D-3}} . \tag{138}$$

In order for the quantum theory to be invariant under large gauge transformations, we must have

$$\delta S = 2\pi i \mathbb{Z} . \tag{139}$$

Equation (138) together with this condition imply the quantization of all elements of $K$. Let us choose, for example, that the only nonvanishing integers in (138) are

$$n^{(1)} = 1, \quad p^{(1)}_{34 \ldots D-3} = 1, \quad \text{and the } (D-3)! \text{ permutations of } p^{(\alpha)}_{34 \ldots D-3} . \tag{140}$$

In this case, (138) becomes

$$
\begin{aligned}
\delta S &= \pi K_{ab} \epsilon_{34 \ldots (D-3) ab} \\
&= 2\pi K_{12} .
\end{aligned}
\tag{141}
$$

Thus, $K_{12}$ must be an integer. By proceeding similarly we get the quantization of all elements of $K$.

The flux quantization (132) also leads properly to the charge quantization. Indeed, by introducing the coupling to a density $-A^{(\alpha)}_0 J^{(\alpha)}_0$, it follows the flux-attachment relation

$$J^{(\alpha)}_0 = \frac{1}{\pi} K_{ab} \mathcal{D}^{(\alpha)}_a A_b . \tag{142}$$

By integrating over $\mathcal{M}^D$, we obtain

$$
\begin{aligned}
Q^{(\alpha)} = \int_{\mathcal{M}^D} J^{(\alpha)}_0 &= \int_{\mathcal{M}^D} \frac{1}{\pi} K_{ab} \mathcal{D}^{(\alpha)}_a A_b \\
&= \frac{1}{2\pi} K_{ab} \int_{\mathcal{M}^D} \left( \mathcal{D}^{(\alpha)}_a A_b - \mathcal{D}^{(\alpha)}_b A_a \right) \\
&= \frac{1}{2\pi} K_{ab} \int_{\mathcal{M}^D} \frac{1}{(D-3)!} \epsilon_{a_1 a_2 \ldots a_{D-3} ab} B^{(\alpha)}_{a_1 a_2 \ldots a_{D-3}} \\
&= \frac{1}{2} K_{ab} \int_{\mathcal{M}^D} \frac{1}{(D-3)!} \epsilon_{a_1 a_2 \ldots a_{D-3} ab} \, p^{(\alpha)}_{a_1 a_2 \ldots a_{D-3}} .
\end{aligned}
\tag{143}
$$

By choosing again the configuration in (140),

$$
\begin{aligned}
Q^{(\alpha)} &= \frac{1}{2}K_{ab}\epsilon_{34\ldots(D-3)ab} \\
&= K_{12},
\end{aligned}
\tag{144}
$$

which is an integer.

## E  $K$-matrix and microscopic theory

The matrix $K$ is determined in three steps: i) we determine its dimensionality; ii) we find constraints on the possible values of the elements; iii) we fix the elements in order to match the physical properties of the lattice model. The steps i) and ii) follow directly from the algebra of the operators in (16). Step iii) is more subtle and follows from the algebra of the ground state holonomy gauge-invariant operators of the effective field theory.

**Step (i): Dimensionality of $K$-matrix.**  In the class of models we have constructed in the manuscript, the dimensionality of the "spin" operator acting at each site is tied to the spatial dimensionality. Indeed,

$$
\begin{aligned}
D = 3 &\quad\Rightarrow\quad \sigma_{i_1} \Leftrightarrow \gamma_{2\times2} \\
D = 5 &\quad\Rightarrow\quad \sigma_{i_1} \otimes \sigma_{i_2} \Leftrightarrow \gamma_{4\times4} \\
D = 2n+1 &\quad\Rightarrow\quad \sigma_{i_1} \otimes \sigma_{i_2} \otimes \cdots \otimes \sigma_{i_n} \Leftrightarrow \gamma_{2^n\times2^n}.
\end{aligned}
\tag{145}
$$

In this way, the algebra of the spin operators in $D$ spatial dimensions can be written in terms of the Clifford algebra of Dirac matrices of dimensionality $2^n \times 2^n$. In this case, we have $2n$ Dirac matrices. Therefore, according to the representation of (16) of the paper, we need $2n$ distinct fields $A$ to reproduce properly the algebra of the Dirac matrices. This fixes the dimensionality of the matrix $K$ to be $2n \times 2n$.

**Step (ii): Elements of the $K$-matrix.**  The possible values of the elements of the $K$-matrix are determined from the algebra of operators at each site. Starting in $D = 3$, with the $K$-matrix given by

$$
K = \begin{pmatrix} 0 & k \\ -k & 0 \end{pmatrix} \quad\text{and}\quad K^{-1} = \begin{pmatrix} 0 & -\frac{1}{k} \\ \frac{1}{k} & 0 \end{pmatrix},
\tag{146}
$$

we have the two operators

$$
\gamma^1 = e^{ikA_2} \quad\text{and}\quad \gamma^2 = e^{-ikA_1}.
\tag{147}
$$

They will anticommute if $k$ is odd. For the $D = 2n+1$ dimensional case, the reasoning is similar. We consider a basis of fields so that the matrix $K$ is block-diagonal (eq.(83) of the manuscript)

$$
QKQ^T = \text{Diag}\left\{ \begin{pmatrix} 0 & k_1 \\ -k_1 & 0 \end{pmatrix}, \begin{pmatrix} 0 & k_2 \\ -k_2 & 0 \end{pmatrix}, \ldots, \begin{pmatrix} 0 & k_n \\ -k_n & 0 \end{pmatrix} \right\}.
\tag{148}
$$

Then, the representation (16) of the paper will provide the properly commutation rules between the operators only if all $k$'s are odd.

**Step (iii): Lattice model and the $K$-matrix entries.** To completely fix the $k_i$'s of the matrix in (148), we need to consider the algebra of gauge-invariant operators. This is worked in detail in Sec.4.4 of the manuscript for the case $D = 3$. The key equation is the algebra given in (71) of the paper. The value of $k_i$ determines the size of the representation of the ground state in a corresponding subdimensional manifold where charge is conserved. In other words, the ground state will possess a $\mathbb{Z}_{2k}$ symmetry that is not present in the lattice model unless $k = 1$. The same reasoning goes in higher dimensions, where we have more pairs of operators satisfying the algebra (71) of the manuscript . Thus, the $K$-matrix corresponding to the lattice model is so that its block-diagonal form has $k_1 = k_2 = \ldots = k_n = 1$. In this sense, the $K$-matrix is determined by the microscopic system since it carries information about the symmetries of the lattice model.

The equation (17) of the manuscript suggest a class of equivalence for the $K$-matrices in our description, in a similar fashion as usual Chern-Simons theories that can lead to the same description with two different $K$-matrices. To make this point clear, consider a redefinition of the basis field in the effective action according to

$$A \to WA \,, \tag{149}$$

where $W$ is a matrix with integer entries. This leads to a theory with new parameters

$$\tilde{K} = W^\top K W \quad \text{and} \quad \tilde{T}^{(I,\alpha)} = W^{-1} T^{(I,\alpha)} \,. \tag{150}$$

Thus, two effective theories with parameters $(K, T)$ and $(\tilde{K}, \tilde{T})$ related through (150), with the matrix $W$ possessing integer entries and $\det W = 1$, describes the same fracton system. Indeed, this implies

$$\mathrm{Pf}(\tilde{K}) = \mathrm{Pf}(W^\top K W) = \mathrm{Pf}(K) \,, \tag{151}$$

and also leaves unchanged the quantization condition (for the principal configuration) given in (21) of the manuscript:

$$t_a^{(I)} (K^\top)_{ab} t_b^{(J)} = \tilde{t}_a^{(I)} (\tilde{K}^\top)_{ab} \tilde{t}_b^{(J)} = 2n^{(IJ)} + (1 - \delta_{IJ}), \quad n^{(IJ)} \in \mathbb{Z}, \tag{152}$$

which is still an even integer if $I = J$ and an odd integer if $I \neq J$.

# F  Conservation Laws and Excitations

## F.1  Case $D = 3$

We will examine here the possible types of defects arising from this model of fractons. Let us consider the case $D = 3$, so that the continuity equation reads

$$
\begin{aligned}
\partial_0 J_0 &= \mathcal{D}_1 J_1 + \mathcal{D}_2 J_2 \\
&= (\partial_x^2 - \partial_z^2) J_1 + (\partial_y^2 - \partial_z^2) J_2 \\
&= (\partial_x + \partial_z)(\partial_x - \partial_z) J_1 + (\partial_y + \partial_z)(\partial_y - \partial_z) J_2 \\
&= \partial_{13}^+ \partial_{13}^- J_1 + \partial_{23}^+ \partial_{23}^- J_2 \,,
\end{aligned}
\tag{153}
$$

where the coordinates are $x_{13}^\pm = x \pm z$ and $x_{23}^\pm = y \pm z$. For simplicity, we have absorbed a factor of $1/4$ in $J_0$.

Let us try, for example, to construct a current corresponding to the creation of a single localized charge. For simplicity, we set $J_2 = 0$. Then, a naive solution of (153) is

$$J_0 = \theta(t)\,\delta(y)\,\delta(x_{13}^+ + a_1)\,\delta(x_{13}^- + b_1) \quad \text{and} \quad J_1 = \delta(t)\,\delta(y)\,\theta(x_{13}^+ + a_1)\,\theta(x_{13}^- + b_1), \tag{154}$$

which corresponds to the creation of a fracton localized at $x = -(a_1 + b_1)/2$, $y = 0$ and $z = (b_1 - a_1)/2$. Notice, however, that this configuration corresponds to a process where charge is not conserved ($Q = 0 \to 1$). Indeed,

$$Q = \int dx\,dy\,dz\,J_0 = \theta(t) \int dy\,dx_{13}^+\,dx_{13}^-\,\delta(y)\delta(x_{13}^+ + a_1)\delta(x_{13}^- + b_1) = \theta(t), \quad \frac{dQ}{dt} = \delta(t).$$
(155)

Consequently, it is not a full-fledged solution of the continuity equation.

We could try to avoid the violation of charge above by inserting a charge of opposite sign in a distinct point, which corresponds

$$
\begin{aligned}
J_0 &= \theta(t)\,\delta(y)\left[\delta(x_{13}^+ + a_1)\,\delta(x_{13}^- + b_1) - \delta(x_{13}^+ + c_1)\,\delta(x_{13}^- + d_1)\right] \\
J_1 &= \delta(t)\,\delta(y)\left[\theta(x_{13}^+ + a_1)\,\theta(x_{13}^- + b_1) - \theta(x_{13}^+ + c_1)\,\theta(x_{13}^- + d_1)\right].
\end{aligned}
$$
(156)

This is compatible with charge conservation in the whole system, $Q = \int dx\,dy\,dz\,J_0$, but we still have to inspect the conservation in the sub-manifolds. Let us consider, for example, the following charge

$$Q^{(++)} = \int dx_{13}^+\,dx_{23}^+\,J_0$$

$$= \theta(t) \int dx_{13}^+\,dx_{23}^+\,\delta(y)\left[\delta(x_{13}^+ + a_1)\,\delta(x_{13}^- + b_1) - \delta(x_{13}^+ + c_1)\,\delta(x_{13}^- + d_1)\right]. \quad (157)$$

We need to be careful in computing the integrals, since the directions $x_{13}^+$ and $x_{23}^+$ are not orthogonal, whereas the directions $x_{13}^+$ and $x_{13}^-$ are orthogonal. This means that we can carry out the integration over $x_{13}^+$ keeping $x_{13}^-$ fixed. Thus, we proceed by integrating over $x_{13}^+$, letting $x_{13}^-$ untouched:

$$Q^{(++)} = \theta(t) \int dx_{23}^+\,\delta(y)\left[\delta(x_{13}^- + b_1) - \delta(x_{13}^- + d_1)\right]. \quad (158)$$

The computation of the remaining integral is a little trick because the directions $x_{13}^+$ and $x_{23}^+$ are not orthogonal, but actually we do not need to compute it to extract useful information. Indeed, this expression shows that in order that the charge $Q^{(++)}$ to be conserved we need to require $b_1 = d_1$. Similarly, by considering the charge

$$
\begin{aligned}
Q^{(-+)} &= \theta(t) \int dx_{13}^-\,dx_{23}^+\,\delta(y)\left[\delta(x_{13}^+ + a_1)\,\delta(x_{13}^- + b_1) - \delta(x_{13}^+ + c_1)\,\delta(x_{13}^- + d_1)\right] \\
&= \theta(t) \int dx_{23}^+\,\delta(y)\left[\delta(x_{13}^+ + a_1) - \delta(x_{13}^+ + c_1)\right],
\end{aligned}
$$
(159)

we see that $a_1 = c_1$ in order that this charge to be conserved. The charges $Q^{(+-)}$ and $Q^{(--)}$ do not provide additional conditions. Taking into account that $a_1 = c_1$ and $b_1 = d_1$ in (156), we see that the density of charges $J_0$ trivially vanishes. In conclusion, the process of creation of a dipole is not compatible with the several conservation laws and, consequently, it is not allowed.

Let us try to find a different type of configuration, which is compatible with the whole set of conservation laws. Consider the density,

$$
\begin{aligned}
J_0 &= \theta(t)\,\delta(y)\left[\delta(x_{13}^+ + a_1)\,\delta(x_{13}^- + b_1) - \delta(x_{13}^+ + c_1)\,\delta(x_{13}^- + d_1)\right. \\
&\quad \left. - \delta(x_{13}^+ + e_1)\,\delta(x_{13}^- + f_1) + \delta(x_{13}^+ + g_1)\,\delta(x_{13}^- + h_1)\right],
\end{aligned}
$$
(160)

and the corresponding flux

$$
\begin{aligned}
J_1 &= \delta(t)\delta(y)\left[\theta(x_{13}^+ + a_1)\theta(x_{13}^- + b_1) - \theta(x_{13}^+ + c_1)\theta(x_{13}^- + d_1)\right. \\
&- \left.\theta(x_{13}^+ + e_1)\theta(x_{13}^- + f_1) + \theta(x_{13}^+ + g_1)\theta(x_{13}^- + h_1)\right],
\end{aligned} \tag{161}
$$

which are compatible with the continuity equation (153). It follows immediately that the charge is conserved in the whole three-dimensional manifold. Next, let us examine the conservation laws in the sub-manifolds. We start with the following charges,

$$
\begin{aligned}
Q^{(+\pm)} &= \int dx_{13}^+ dx_{23}^\pm J_0 \\
&= \theta(t)\int dx_{23}^\pm \left[\delta(x_{13}^- + b_1) - \delta(x_{13}^- + d_1) - \delta(x_{13}^- + f_1) + \delta(x_{13}^- + h_1)\right]. \tag{162}
\end{aligned}
$$

We have two possibilities ensuring charge conservation:

$$
\begin{aligned}
&i)\ b_1 = d_1 \quad \text{and} \quad f_1 = h_1 \\
&ii)\ b_1 = f_1 \quad \text{and} \quad d_1 = h_1. \tag{163}
\end{aligned}
$$

Similarly, the remaining charges are

$$
\begin{aligned}
Q^{(-\pm)} &= \int dx_{13}^- dx_{23}^\pm J_0 \\
&= \theta(t)\int dx_{23}^\pm \left[\delta(x_{13}^+ + a_1) - \delta(x_{13}^+ + c_1) - \delta(x_{13}^+ + e_1) + \delta(x_{13}^+ + g_1)\right], \tag{164}
\end{aligned}
$$

which leads also to two possibilities

$$
\begin{aligned}
&i)\ a_1 = c_1 \quad \text{and} \quad e_1 = g_1 \\
&ii)\ a_1 = e_1 \quad \text{and} \quad c_1 = g_1. \tag{165}
\end{aligned}
$$

From these possibilities, it is clear that if we select choice $i)$ of (163) and $i)$ of (165), or $ii)$ of (163) and $ii)$ of (165), the density in (160) will trivially vanish. However, we obtain a non-vanishing density if we choose crosswise $i)/ii)$ of (163) and $ii)/i)$ of (165). Let us choose, say, $i)$ from (163) and $ii)$ from (165). In this case, the density becomes

$$
\begin{aligned}
J_0 &= \theta(t)\delta(y)\left[\delta(x_{13}^+ + a_1)\delta(x_{13}^- + b_1) - \delta(x_{13}^+ + c_1)\delta(x_{13}^- + b_1)\right. \\
&- \left.\delta(x_{13}^+ + a_1)\delta(x_{13}^- + f_1) + \delta(x_{13}^+ + c_1)\delta(x_{13}^- + f_1)\right], \tag{166}
\end{aligned}
$$

which corresponds to the creation of four charges at the following positions:

$$
\begin{aligned}
\text{charge} \quad q_1 &= + \quad \Rightarrow (x,z) = \left(-\frac{a_1 + b_1}{2}, \frac{b_1 - a_1}{2}\right) \\
\text{charge} \quad q_2 &= - \quad \Rightarrow (x,z) = \left(-\frac{c_1 + b_1}{2}, \frac{b_1 - c_1}{2}\right) \\
\text{charge} \quad q_3 &= - \quad \Rightarrow (x,z) = \left(-\frac{a_1 + f_1}{2}, \frac{f_1 - a_1}{2}\right) \\
\text{charge} \quad q_4 &= + \quad \Rightarrow (x,z) = \left(-\frac{c_1 + f_1}{2}, \frac{f_1 - c_1}{2}\right). \tag{167}
\end{aligned}
$$

Let $d(q_i, q_j)$ be the distance between two charges. The above expressions ensure that $d(q_1, q_2) = d(q_3, q_4)$ and $d(q_1, q_3) = d(q_2, q_4)$, which physically means that the sum of all

dipole moments of the configuration vanishes (see figure 2). This guarantees conservation of dipole moment, which is a consequence of the conservation of charges in sub-manifolds (planes). In fact, charge conservation along a plane implies that the dipole moment perpendicular to the plane is conserved.

We see that the location of the four charges in (167) are specified by the set of arbitrary points $a_1, b_1, c_1, f_1$. By varying the values of these points we change both the size of the dipoles and their positions, in a way that preserves the structure depicted in figure 2, i.e., the charges are always localized at the corners of a parallelogram. Physically, this means that the dipoles can move freely in the system, but cannot be created or annihilated (remembering our previous discussion, the creation of a single dipole is not compatible with all the conservation laws).

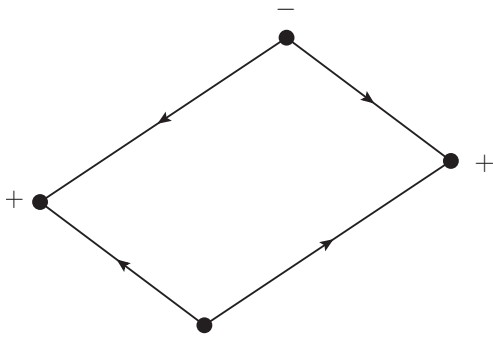

Figure 2: A generic four-charge configuration in the plane $x - z$, as given in (167). It is clear from this figure that the total dipole vanishes.

Before closing, it is instructive to consider a simple symmetric choice, $a_1 = b_1 = a$ and $c_1 = f_1 = -a$. In this case, the density reduces to

$$
\begin{aligned}
J_0 &= \theta(t)\,\delta(y)\big[\delta(x_{13}^+ + a)\,\delta(x_{13}^- + a) - \delta(x_{13}^+ - a)\,\delta(x_{13}^- + a) \\
&- \delta(x_{13}^+ + a)\,\delta(x_{13}^- - a) + \delta(x_{13}^+ - a)\,\delta(x_{13}^- - a)\big],
\end{aligned}
\tag{168}
$$

while the flux can be written as

$$
\begin{aligned}
J_1 &= \delta(t)\,\delta(y)\big[\theta(x_{13}^+ + a)\,\theta(x_{13}^- + a) - \theta(x_{13}^+ - a)\,\theta(x_{13}^- + a) \\
&- \theta(x_{13}^+ + a)\,\theta(x_{13}^- - a) + \theta(x_{13}^+ - a)\,\theta(x_{13}^- - a)\big] \\
&= \delta(t)\,\delta(y)\,\theta(a + x_{13}^+)\,\theta(a - x_{13}^+)\,\theta(a + x_{13}^-)\,\theta(a - x_{13}^-),
\end{aligned}
\tag{169}
$$

where to write in terms of a single term we have used the property $\theta(x) + \theta(-x) = 1$. The density $J_0$ describes the creation of a set of four charges located at the points

$$
\begin{aligned}
x = \pm a,\ y = 0,\ z = 0 \quad &\Rightarrow \quad \text{positive charges} \\
x = 0,\ y = 0,\ z = \pm a \quad &\Rightarrow \quad \text{negative charges}.
\end{aligned}
\tag{170}
$$

This configuration is depicted in figure 3.

## F.2 Case $D = 5$

In this case we have two conservation laws given by (51),

$$
\partial_0 J_0^{(1)} = D_1 J_1^{(1)} + D_2 J_2^{(1)} + D_3 J_3^{(1)} + D_4 J_4^{(1)},
\tag{171}
$$

and

$$
\partial_0 J_0^{(2)} = D_1 J_1^{(2)} + D_2 J_2^{(2)} + D_3 J_3^{(2)} + D_4 J_4^{(2)},
\tag{172}
$$

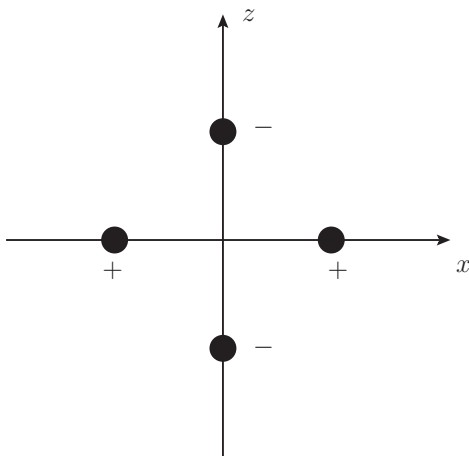

Figure 3: Charge configuration corresponding to the process described by the density in (156).

where $J_I^{(\alpha)} = T_a^{(I,\alpha)} J_a$. For $\alpha = 1$ and the canonical form of $t$'s,

$$J_I^{(1)} = t_a^{(I)} J_a = J_I \,, \tag{173}$$

whereas for $\alpha = 2$ it is convenient to write all components explicitly,

$$
\begin{aligned}
J_1^{(2)} &= T_1^{(1,2)} J_1 + T_2^{(1,2)} J_2 + T_3^{(1,2)} J_3 + T_4^{(1,2)} J_4 \\
J_2^{(2)} &= T_1^{(2,2)} J_1 + T_2^{(2,2)} J_2 + T_3^{(2,2)} J_3 + T_4^{(2,2)} J_4 \\
J_3^{(2)} &= T_1^{(3,2)} J_1 + T_2^{(3,2)} J_2 + T_3^{(3,2)} J_3 + T_4^{(3,2)} J_4 \\
J_4^{(2)} &= T_1^{(4,2)} J_1 + T_2^{(4,2)} J_2 + T_3^{(4,2)} J_3 + T_4^{(4,2)} J_4 \,.
\end{aligned}
\tag{174}
$$

Our goal here is the following. We will create an elementary excitation in the lattice model and then we want to understand how this is reproduced from the conservation laws above. Let us consider the case $D = 5$ given in (14). The application of the local operator $\gamma^1 \gamma^2$ in a particular site of the even sub-lattice creates four defects of type $\alpha = 1$ in the plane $x_1 - x_2$ and four defects of type $\alpha = 2$ in the plane $x_3 - x_4$. Now let us see how this arises from the point of view of the conservation laws.

These excitations can be reproduced with $J_3 = J_4 = 0$ and $J_2 = -J_1$, so that (171) becomes

$$
\begin{aligned}
\partial_0 J_0^{(1)} &= (D_1 - D_2) J_1 \\
&= (\partial_1^2 - \partial_2^2) J_1 \\
&= \partial_{12}^+ \partial_{12}^- J_1 \,.
\end{aligned}
\tag{175}
$$

To construct the currents for $\alpha = 2$ we can read the vectors $T$ from (122). We remember, however, that their components are defined only mod 2, so that it is convenient to choose

$$
\begin{aligned}
J_1^{(2)} &= J_2^{(2)} = 0 \\
J_3^{(2)} &= -J_4^{(2)} = J_1 \,.
\end{aligned}
\tag{176}
$$

With this, the conservation law (172) becomes

$$\partial_0 J_0^{(2)} = \partial_{34}^+ \partial_{34}^- J_1 \,. \tag{177}$$

We can construct a current $J_1$ that creates excitations simultaneously in planes $x_1 - x_2$ and $x_3 - x_4$ by using the four-charge configurations of the case $D = 3$ (161) with the positions

of the charges subject to (167), since these configurations also live in planes. In this way, we write the generalization for the five-dimensional case as

$$J_1 = \delta(t)\,\delta(x_3)\,\delta(x_4)\,\delta(x_5)\,\Theta(x_{12}^+, x_{12}^-) + \delta(t)\,\delta(x_1)\,\delta(x_2)\,\delta(x_5)\,\Theta(x_{34}^+, x_{34}^-),\tag{178}$$

where $\Theta(x_{12}^+, x_{12}^-)$ is defined as the $\theta$-dependent part of (161):

$$\begin{aligned}
\Theta(x_{12}^+, x_{12}^-) &\equiv& \theta(x_{12}^+ + a_1)\,\theta(x_{12}^- + b_1) - \theta(x_{12}^+ + c_1)\,\theta(x_{12}^- + b_1)\\
&-& \theta(x_{12}^+ + a_1)\,\theta(x_{12}^- + f_1) + \theta(x_{12}^+ + c_1)\,\theta(x_{12}^- + f_1).
\end{aligned}\tag{179}$$

Plugging $J_1$ in (175) and (177) gives the densities

$$\begin{aligned}
J_0^{(1)} &=& \theta(t)\,\delta(x_3)\,\delta(x_4)\,\delta(x_5)\,\Delta(x_{12}^+, x_{12}^-)\\
&+& \theta(t)\,\delta(x_5)\,\Theta(x_{34}^+, x_{34}^-)\,\partial_{12}^+\,\partial_{12}^-\,\delta(x_1)\,\delta(x_2),
\end{aligned}\tag{180}$$

and

$$\begin{aligned}
J_0^{(2)} &=& \theta(t)\,\delta(x_1)\,\delta(x_2)\,\delta(x_5)\,\Delta(x_{34}^+, x_{34}^-)\\
&+& \theta(t)\,\delta(x_5)\,\Theta(x_{12}^+, x_{12}^-)\,\partial_{34}^+\,\partial_{34}^-\,\delta(x_3)\,\delta(x_4),
\end{aligned}\tag{181}$$

where

$$\begin{aligned}
\Delta(x_{12}^+, x_{12}^-) &\equiv& \partial_{12}^+\,\partial_{12}^-\,\Theta(x_{12}^+, x_{12}^-)\\
&=& \delta(x_{12}^+ + a_1)\,\delta(x_{12}^- + b_1) - \delta(x_{12}^+ + c_1)\,\delta(x_{12}^- + b_1)\\
&-& \delta(x_{12}^+ + a_1)\,\delta(x_{12}^- + f_1) + \delta(x_{12}^+ + c_1)\,\delta(x_{12}^- + f_1).
\end{aligned}\tag{182}$$

There are some important points to notice in the densities $J_0^{(1)}$ and $J_0^{(2)}$. The terms in the first lines of both (180) and (181) correspond indeed to four-charge configurations with vanishing total dipole, like in the case $D = 3$. But now, we have additional terms in the second lines. However, such terms do not affect the physical charge and can be absorbed in a redefinition of the currents. Indeed, we can define

$$\tilde{J}_0^{(1)} \equiv J_0^{(1)} - \Omega_0^{(1)} \quad \text{and} \quad \tilde{J}_1^{(1)} \equiv J_1 - \Omega_1^{(1)}.\tag{183}$$

with similar definitions for the currents of $\alpha = 2$, i.e., $\tilde{J}_0^{(2)} \equiv J_0^{(2)} - \Omega_0^{(2)}$ and $J_1^{(2)} \equiv J_1 - \Omega_1^{(2)}$. If $\Omega_0^{(1)}$ and $\Omega_1^{(1)}$ satisfy

$$\partial_0\,\Omega_0^{(1)} = \partial_{12}^+\,\partial_{12}^-\,\Omega_1^{(1)},\tag{184}$$

and

$$\int dx_{15}^{\sigma_1}\,dx_{25}^{\sigma_2}\,dx_{35}^{\sigma_3}\,dx_{45}^{\sigma_4}\,\Omega_0^{(1)} = 0,\tag{185}$$

then the two currents $(J_0^{(1)}, J_1)$ and $(\tilde{J}_0^{(1)}, \tilde{J}_1^{(1)})$ describe the same physical situation, since the redefined currents also satisfy

$$\partial_0\,\tilde{J}_0^{(1)} = \partial_{12}^+\,\partial_{12}^-\,\tilde{J}_1^{(1)},\tag{186}$$

and

$$\int dx_{15}^{\sigma_1}\,dx_{25}^{\sigma_2}\,dx_{35}^{\sigma_3}\,dx_{45}^{\sigma_4}\,\tilde{J}_0^{(1)} = \int dx_{15}^{\sigma_1}\,dx_{25}^{\sigma_2}\,dx_{35}^{\sigma_3}\,dx_{45}^{\sigma_4}\,J_0^{(1)}.\tag{187}$$

From equations (178) and (180) we see that if we set,

$$\Omega_0^{(1)} = \theta(t)\,\delta(x_5)\,\Theta(x_{34}^+, x_{34}^-)\partial_{12}^+\,\partial_{12}^-\,\delta(x_1)\,\delta(x_2),\tag{188}$$

and

$$\Omega_1^{(1)} = \delta(t)\,\delta(x_1)\,\delta(x_2)\,\delta(x_5)\,\Theta(x_{34}^+, x_{34}^-)\,, \tag{189}$$

then the condition (184) is immediately satisfied.

Next, let us consider (185),

$$\int dx_{15}^{\sigma_1}\,dx_{25}^{\sigma_2}\,dx_{35}^{\sigma_3}\,dx_{45}^{\sigma_4}\,\theta(t)\,\delta(x_5)\,\Theta(x_{34}^+, x_{34}^-)(\partial_1^2 - \partial_2^2)\,\delta(x_1)\,\delta(x_2)\,. \tag{190}$$

This term vanishes identically. To see this, we notice that as $x_{15}^{\sigma_1} = x_1 + \sigma_1 x_5$ and $x_{25}^{\sigma_2} = x_2 + \sigma_2 x_5$, with $\sigma_1, \sigma_2 = \pm$, under the change of variables

$$x_1 \to \sigma_1 \sigma_2 x_2 \quad \text{and} \quad x_2 \to \sigma_1 \sigma_2 x_1\,, \tag{191}$$

the integration measure transforms as

$$dx_{15}^{\sigma_1} \to \sigma_1 \sigma_2\, dx_{25}^{\sigma_2} \quad \text{and} \quad dx_{25}^{\sigma_2} \to \sigma_1 \sigma_2\, dx_{15}^{\sigma_1}\,, \tag{192}$$

so that $dx_{15}^{\sigma_1}\,dx_{25}^{\sigma_2}$ is invariant (even). On the other hand, the integrand $(\partial_1^2 - \partial_2^2)\delta(x_1)\delta(x_2)$ is odd and hence the integral vanishes. Therefore, we can construct a redefined density simply as

$$\tilde{J}_0^{(1)} = \theta(t)\,\delta(x_3)\,\delta(x_4)\,\delta(x_5)\,\Delta(x_{12}^+, x_{12}^-)\,, \tag{193}$$

which corresponds to the creation of a four-charge configuration in the plane $x_1 - x_2$. We can proceed in the same way for the density in (181), and define

$$\tilde{J}_0^{(2)} = \theta(t)\,\delta(x_1)\,\delta(x_2)\,\delta(x_5)\,\Delta(x_{34}^+, x_{34}^-)\,. \tag{194}$$

It remains to show that these densities satisfy the requirement of charge conservation. This is not immediate because the four-charge configurations $\Delta(x_{12}^+, x_{12}^-)$ and $\Delta(x_{34}^+, x_{34}^-)$ involve directions which are not appearing in the integration measure (53). For example, consider the charge

$$\begin{aligned} Q_{(\sigma_1,\sigma_2,\sigma_3,\sigma_4)}^{(1)} &= \int dx_{15}^{\sigma_1}\,dx_{25}^{\sigma_2}\,dx_{35}^{\sigma_3}\,dx_{45}^{\sigma_4}\,\tilde{J}_0^1 \\ &= \theta(t)\int dx_{15}^{\sigma_1}\,dx_{25}^{\sigma_2}\,dx_{35}^{\sigma_3}\,dx_{45}^{\sigma_4}\,\delta(x_3)\,\delta(x_4)\,\delta(x_5)\,\Delta(x_{12}^+, x_{12}^-)\,. \end{aligned} \tag{195}$$

We have to change the integration from $x_{15}^{\pm}$ to $x_{12}^{\pm}$, since we know that $\int dx_{12}^{\sigma_1}\,\Delta(x_{12}^+, x_{12}^-) = 0$. This can be done in the following way:

$$\begin{aligned} x_{15}^{\sigma_1} &= x_1 + \sigma_1 x_5\,, \\ &= x_1 + \tilde{\sigma}_1 x_2 + \sigma_1 x_5 - \tilde{\sigma}_1 x_2\,, \\ &= x_{12}^{\tilde{\sigma}_1} - \tilde{\sigma}_1 x_{25}^{-\tilde{\sigma}_1\sigma_1}\,. \end{aligned} \tag{196}$$

As the change from $x_{15}^{\sigma_1}$ to $x_{12}^{\tilde{\sigma}_1}$ involves $x_{25}^{\pm}$, we have to ensure that the coordinate appearing in this expression is the opposite to the coordinate in the integration measure $dx_{25}^{\sigma_2}$, since the directions $x_{25}^+$ and $x_{25}^-$ are orthogonal. To this, we just need to set $\tilde{\sigma}_1 = \sigma_1\sigma_2$,

$$x_{15}^{\sigma_1} = x_{12}^{\sigma_1\sigma_2} - \sigma_1\sigma_2 x_{25}^{-\sigma_2}\,. \tag{197}$$

Therefore, as $x_{25}^{-\sigma_2}$ is fixed in the integration along the direction $x_{25}^{\sigma_2}$, we can directly write $dx_{15}^{\sigma_1} = dx_{12}^{\sigma_1\sigma_2}$, so that

$$Q_{(\sigma_1,\sigma_2,\sigma_3,\sigma_4)}^{(1)} = \theta(t)\int dx_{12}^{\sigma_1}\,dx_{25}^{\sigma_2}\,dx_{35}^{\sigma_3}\,dx_{45}^{\sigma_4}\,\delta(x_3)\,\delta(x_4)\,\delta(x_5)\,\Delta(x_{12}^+, x_{12}^-) = 0\,, \tag{198}$$

where we have renamed $\sigma_1\sigma_2 \to \sigma_1$. The same reasoning can be done with the charges associated with the density $\tilde{J}_0^{(2)}$.

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
