# Peer review of "Lattice Clifford fractons and their Chern-Simons-like theory"

_SciPost Physics, doi:SciPost Phys. Core 4, 012 (2021)_

## Round 2 · Referee Report · Anonymous (Referee 1) · 2020-8-22

Strengths

1) This work is thorough, very timely, and all the ideas and results are presented in a clear and transparent manner.

2) The identification of continuum descriptions of fracton models on the lattice in terms of Chern-Simons-like theories gives a good way to understand the connection between the well-studied fracton lattice models and field theories.

3) The connection to hierarchical quantum Hall states is interesting and well-explained.

Weaknesses

The authors could consider adding short discussions addressing the following:

1) It would be interesting to mention how this can be connected to the foliation paradigm present in the fracton literature.

2) How does this formalism capture the non-trivial statistics between the different subdimensional particles?

3) Can this formalism be extended to SSPT models on lattices with more exotic geometries, such as hyperbolic geometries? In this case, the group of translations results in a non-abelian group, which could potentially allow for more interesting physics.

Report

This is a well-written and timely paper invoking a
novel approach to address an important conceptual problem related to connecting lattice fracton models with field theory descriptions. The paper is of good quality, and will make a fine contribution to SciPost.

Requested changes

Below, I list some optional additions (same as in the "weaknesses" section) that the authors could consider:

1) It would be interesting to mention how this can be connected to the foliation paradigm present in the fracton literature.

2) How does this formalism capture the non-trivial statistics between the different subdimensional particles?

3) Can this formalism be extended to SSPT models on lattices with more exotic geometries, such as hyperbolic geometries. In this case, the group of translations results in a non-abelian group, which could potentially allow for more interesting physics.

  • validity: high
  • significance: good
  • originality: good
  • clarity: high
  • formatting: good
  • grammar: good

Author:  Weslei Fontana  on 2020-12-18  [id 1084]

(in reply to Report 1 on 2020-08-22)
Category:
answer to question

The referee writes

It would be interesting to mention how this can be connected to the foliation paradigm present in the fracton literature.

Our response Indeed as the referee suggested there should be a connection between our construction and the foliation paradigm of Shirley, Slagle and Chen. In the D=3 case it was shown (https://arxiv.org/abs/1812.01613v3) and (https://doi.org/10.1016/j.aop.2019.167922) that the model has 4 foliations which emerge as the number of submanifolds with charge conservation. In our construction for the D-dimensional case we find $2^{D-1} submanifolds where charge is conserved. For this case the exact foliation structure is not obvious to us, but we expect that the number of foliations should increase as a function of D. Determining the exact number of foliations for the D-dimensional problem it is certainly an interesting open question.

The referee writes

How does this formalism capture the non-trivial statistics between the different subdimensional particles?

Our response The standard definition of braiding quasiparticles requires full mobility of excitations, which is not the case for fracton models. As it is stressed in the main text, the only mobile excitations of the Chamon model are lineons, quasiparticles that moves strictly along lines on the submanifolds with charge conservation. This mobility restriction makes it unnatural to speak of braiding as is known in the standard literature. A similar reasoning applies to the fractons with fully immobile quasiparticles.

Some authors have proposed alternative ways to define the statistics of excitations in fracton phases. For instance, in (https://doi.org/10.1103/PhysRevB.100.195136) the authors build a fusion theory for fractons that incorporates the non-trivial mobility of the excitations and from the fusion theory they built a notion of statistics by means of local creation and annihilation processes. These notions can, in principle, be extended to the higher dimensional fractons we constructed in our paper.

The referee writes

Can this formalism be extended to SSPT models on lattices with more exotic geometries, such as hyperbolic geometries. In this case, the group of translations results in a non-abelian group, which could potentially allow for more interesting physics.

Our response Our construction is generic in that it applies to systems whose microscopic Hamiltonian is a sum of commuting projectors built from tensor products of spin-1/2 operators. With this in mind, we expect that this formalism can be applied to SSPT models as long as they satisfy these criteria, although each particular model will present its own subtleties and some caution is needed.

Our construction does depend on the lattice structure, so if we apply the formalism for models wich are embedded in lattices with exotic geometries, this information will be carried in a non-trivial way to the T-vectors.

---

## Round 2 · Referee Report · Anonymous (Referee 2) · 2021-3-8

Strengths

The authors discuss a novel class of gapped fracton models in all odd spatial dimensions, which includes the Chamon model as the simplest case, and propose the low energy continuum field theory description.

Weaknesses

(1) In various discussions it is not clear which quantities are used to define the microscopic lattice models, and which are derived quantities in the continuum field theory. See my comments in Requested Changes.

(2) Various global aspects of the continuum field theory are not discussed in detail.

Report

While the results are interesting, various crucial aspects of the continuum field theory are not addressed, and many other discussions could still be improved. I would request a major revision before it is considered for publication.

Requested changes

Major changes:

(1) If I understand correctly, the T-vectors are microscopic data that enter into the definition of the lattice Hamiltonian as in equation (8). On the other hand, the matrix $K_{ab}$ are coefficients of the continuum Lagrangian. Starting from a given microscopic lattice model, there should be an unambiguous way to determine the macroscopic parameters. In section III and in appendix B, it is not clear how the matrix $K_{ab}$ is determined from the microscopic data.

Relatedly, I am confused why the authors say in the Final Remarks that "The details about an specific lattice model enter vial this matrix K...". The matrix $K_{ab}$ is introduced after the authors have defined the lattice model in equation (8).

(2) In ordinary abelian Chern-Simons theory, two different K matrices can define the same continuum field theory. What's the corresponding identification for $K_{ab}$ in the current context?

(3) In the discussion of the continuum field theory, it is not clear whether the gauge fields $A$ are $U(1)$-valued or $\mathbb{R}$-valued. In the former case, there is no need for the integer field $m^{(\alpha)}$. In the latter case, the $\mathbb{R}$-valued gauge fields $A$ are further subject to an integer gauge transformation, and the field $m^{(\alpha)}$ is the corresponding integer gauge field.

The authors are not consistent in distinguishing these two presentations. In (38), it seems that the fields $A$ are $\mathbb{R}$-valued, but then in (40) and in (43), they become $U(1)$-valued and there is no integer field $m^{(\alpha)}$. The authors could have worked with $U(1)$-valued gauge fields throughout, and comment that different charge sectors arise from nontrivial twisted sectors of the $U(1)$-valued gauge fields (rather from an independent integer field $m^{(\alpha)}$), as is standard in field theory.

(4) Starting from a microscopic lattice model, it is conceivable that the $K_{ab}$ matrix is naturally quantized in some way. This is perhaps the perspective in section IV.A. However, if one starts from the continuum field theory (38) as given, it is not clear why the matrix $K_{ab}$ has to be quantized. In ordinary Chern-Simons theory, the K matrix is quantized by the large gauge transformations. The authors should discuss what large gauge transformations quantize their $K_{ab}$.

Minor changes:

(1) The authors are not consistent with their convention for the space and spacetime dimensions. For example, in the last sentence on page 2, both "3D" and "(2+1)-dimensional" are used in the same sentence. Also, in the abstract, by "(D+1) effective theory", I assume the authors mean "an effective theory in (D+1) spacetime dimensions."

(2) Perhaps the authors can provide a more intuitive explanation why the number of qubits on each site $n$ is correlated with the spatial dimension $D=2n+1$. This is not motivated in the introduction on page 3.

(3) The indices $a,b$ are not defined in equation (1) (although it is defined later).

(4) Above equation (7), $\Lambda_o$ is not a sublattice: the sum of two odd vectors is even.

(5) Below (43), "be an integer odd".

(6) In the Final Remarks, the authors comment that the same construction applies to the Haah code, which is in 3 spatial dimensions. However, there are 2 qubits per site in the Haah code, in which case the authors' general construction would yield a fracon model in $2\times 2+1=5$ spatial dimensions. Perhaps the authors can clarify this point.

---

## Round 4 · Author Response

Dear Editor,
We are grateful to the referee for the pertinent criticisms. Following her/his comments, we revised the manuscript. We have addressed all the requested changes.
Sincerely,
The Authors
We are grateful to the referee for the pertinent criticisms. Following her/his comments, we revised the manuscript. We have addressed all the requested changes.
Sincerely,
The Authors

---

## Round 4 · List of Changes

*We added to the manuscript an additional appendix (Appendix E) with the discussion on how to determine the elements of the K-matrix;
*We have also included in the additional Appendix E the discussion of how theories with different K-matrices are equivalent,
*We have added to the manuscript equations (22) and (23) and a discussion about the compactification of the fields;
*We have included in the manuscript a discussion about the level quantization in Sec. IV A, and
also a new appendix (Appendix D) with further details of the calculation;
*In addition to the major changes required by the referee, we have also addressed all the listed
minor changes.
We thank the referee for the constructive and valuable criticism.
*We have also included in the additional Appendix E the discussion of how theories with different K-matrices are equivalent,
*We have added to the manuscript equations (22) and (23) and a discussion about the compactification of the fields;
*We have included in the manuscript a discussion about the level quantization in Sec. IV A, and
also a new appendix (Appendix D) with further details of the calculation;
*In addition to the major changes required by the referee, we have also addressed all the listed
minor changes.
We thank the referee for the constructive and valuable criticism.

---

## Editorial Decision

published